# FORKS: Fast Second-Order Online Kernel Learning using Incremental Sketching

## Abstract

Online Kernel Learning (OKL) has attracted considerable research interest due to its promising predictive performance. Second-order methods are particularly appealing for OKL as they often offer substantial improvements in regret guarantees. However, existing approaches like PROS-N-KONS suffer from at least quadratic time complexity with respect to the budget, rendering them unsuitable for meeting the real-time demands of large-scale online learning. Additionally, current OKL methods are typically prone to concept drifting in data streams, making them vulnerable in adversarial environments. To address these issues, we introduce FORKS, a fast incremental sketching approach for second-order online kernel learning. FORKS maintains an efficient time-varying explicit feature mapping that enables rapid updates and decomposition of sketches using incremental sketching techniques. Theoretical analysis demonstrates that FORKS achieves a logarithmic regret guarantee, on par with other second-order approaches, while maintaining a linear time complexity w.r.t. the budget. We validate the performance of FORKS through extensive experiments conducted on real-world datasets, demonstrating its superior scalability and robustness against adversarial attacks.

## 1 Introduction

The objective of online learning is to efficiently and effectively update hypotheses in a data stream environment, where the processes of training and testing are intermixed (Shalev-Shwartz, 2011). A popular online learning algorithm is Online Gradient Descent (OGD), which aims to minimize the loss function by iteratively adjusting the parameters in the direction of the negative gradient of the function (Zinkevich, 2003). However, OGD only uses the linear combination of input features, which makes it susceptible to challenges posed by nonlinear problems. To overcome this limitation, Online Kernel Learning (OKL) generates a feature mapping from the input space to a high-dimensional reproducing kernel Hilbert space (RKHS) in order to effectively handle nonlinear learning tasks (Kivinen et al., 2004; Lu et al., 2016b; Singh et al., 2012; Sahoo et al., 2019; Hu et al., 2015).

OKL can be categorized into first-order and second-order approaches. To achieve logarithmic regret with respect to the number of rounds, the first-order methods require the assumption that the loss function exhibits strong convexity. However, this assumption is unrealistic for most loss functions. In contrast, second-order OKL approaches can achieve logarithmic regret without requiring strong convexity along all directions, enabling them to learn the optimal hypothesis more efficiently. Currently, the only approximate second-order OKL approaches known to achieve logarithmic regret are SKETCHED-KONS and PROS-N-KONS (Calandriello et al., 2017b;a). Both approaches are built upon the exact second-order optimization method Online Newton Step (ONS). Besides, these methods rely on online sampling techniques, which involve the incremental construction of non-uniform sampling distributions, rendering a significant cost of updates.

However, existing second-order approaches, including PROS-N-KONS, suffer from two notable challenges. First, PROS-N-KONS exhibits at least quadratic time complexity with respect to the budget, making it unsuitable for large-scale online learning tasks that require real-time processing. Although several existing first-order methods have successfully reduced their running time to linear complexity by leveraging function approximation techniques, the extension of such techniques to second-order approaches requires further exploration (Cavallanti et al., 2007; Wang & Vucetic, 2010; Zhao et al., 2012; Lu et al., 2016a). Second, most existing first- and second-order approaches are prone to

concept drifting in data streams, making them susceptible to adversarial environments (Zhang & Liao, 2019). Zhang & Liao (2019) use rank-1 modifications to update incremental randomized sketches and create a time-varying explicit feature mapping to demonstrate better learning performance in terms of accuracy and efficiency, even in adversarial environments. Nevertheless, their approach is limited to first-order gradient descent due to high computational complexity and susceptibility to error accumulation. Motivated by these challenges, our work aims to address the following question: *Can we construct a second-order online kernel learning algorithm with efficient and effective updates?* In this paper, we provide an affirmative answer by introducing a fast incremental sketching approach for second-order online kernel learning and a novel decomposition method tailored to sketch updates. Our contributions can be summarized as follows:

• We propose FORKS, a fast and effective second-order online kernel learning method that can be generalized to both regression and classification tasks. FORKS maintains incremental randomized sketches using efficient low-rank modifications and constructs an effective time-varying explicit feature mapping. We provide a detailed theoretical analysis to illustrate the advantages of FORKS, including having linear time complexity w.r.t. the budget, and enjoying a logarithmic regret bound.

• We propose TISVD, a novel Truncated Incremental Singular Value Decomposition adapting to matrix decomposition problems in online learning environments. We theoretically compare the time complexity between TISVD and the original truncated low-rank SVD, confirming that FORKS with TISVD is computationally more efficient without compromising prediction performance.

• We conduct an extensive experimental study to demonstrate the superior performance of FORKS on both adversarial and real-world datasets while maintaining practical computational complexity. Furthermore, we validate the robustness and scalability of FORKS on large-scale datasets.

## 2 PRELIMINARIES

**Notations.** Let $[n] = \{1, 2, \ldots, n\}$, upper-case bold letters (e.g., $\boldsymbol{A}$) represent matrix and lower-case bold letters (e.g., $\boldsymbol{a}$) represent vectors. We denote by $\boldsymbol{A}_{i*}$ and $\boldsymbol{A}_{*j}$ the $i$-th row and $j$-th column of matrix $\boldsymbol{A}$, $\boldsymbol{A}^\dagger$ the Moore-Penrose pseudoinverse of $\boldsymbol{A}$, $\|\boldsymbol{A}\|_2$ and $\|\boldsymbol{A}\|_F$ the spectral and Frobenius norms of $\boldsymbol{A}$. Let $\mathcal{S} = \{(\boldsymbol{x}_t, y_t)\}_{t=1}^T \subseteq (\mathcal{X} \times \mathcal{Y})^T$ be the data stream of $T$ instances, where $\boldsymbol{x}_t \in \mathbb{R}^D$. We use $\boldsymbol{A} = \boldsymbol{U}\boldsymbol{\Sigma}\boldsymbol{V}^\top$ to represent the SVD of $\boldsymbol{A}$, where $\boldsymbol{U}, \boldsymbol{V}$ denote the left and right matrices of singular vectors and $\boldsymbol{\Sigma} = \text{diag}[\lambda_1, ..., \lambda_n]$ is the diagonal matrix of singular values.

**Online Kernel Learning.** We denote the kernel function by $\kappa : \mathcal{X} \times \mathcal{X} \to \mathbb{R}$ and the corresponding kernel matrix by $\boldsymbol{K} = (\kappa(\boldsymbol{x}_i, \boldsymbol{x}_j))$. Let $\mathcal{H}_\kappa$ be the RKHS induced by $\kappa$, and the corresponding feature mapping $\boldsymbol{\phi} : \mathcal{X} \to \mathcal{H}_\kappa$. In this case, the kernel function can be represented as the inner product $\kappa(\boldsymbol{x}_i, \boldsymbol{x}_j) = \boldsymbol{\phi}^\top(\boldsymbol{x}_i)\boldsymbol{\phi}(\boldsymbol{x}_j)$. We consider the online classification setting, i.e., $\mathcal{Y} = \{-1, 1\}$. Given a data stream $\mathcal{S}$ and a convex loss function $\ell$, we define the hypothesis by $f_t$ at round $t$. When a new example $\boldsymbol{x}_t$ arrives, the hypothesis predicts label $\hat{y}_t$ using $f_t$. Then, the hypothesis incurs loss $\ell_t(f_t(\boldsymbol{x}_t)) := \ell(f_t(\boldsymbol{x}_t), y_t)$ and updates its model parameters. The goal of an online learning algorithm is to bound the cumulative regret between the hypothesis and an optimal hypothesis $f^*$ in hindsight. The regret can be defined as $\text{Reg}_T(f^*) = \sum_{t=1}^T [\ell_t(f_t) - \ell_t(f^*)]$, where $f^* = \arg\min_{f \in \mathcal{H}_\kappa} \sum_{t=1}^T \ell_t(f)$.

## 3 FORKS: THE PROPOSED ALGORITHM

While certain endeavors have been undertaken to apply the sketching approach to OKL (Lu et al., 2016a; Cavallanti et al., 2007; Zhang & Liao, 2019), it still harbors inherent limitations that hinder its scalability to second-order methods. First, imposing the decomposition operation on the sketch is inefficient, leading to expensive computational costs when updating the feature mapping. Second, directly performing second-order optimization in the original RKHS with implicit feature mapping has high computational complexity due to the growing size of the Hessian matrix. To address these limitations, we propose an efficient second-order online kernel learning procedure, named **FORKS**.

### 3.1 CONSTRUCTING EXPLICIT FEATURE MAPPING WITH RANDOMIZED SKETCHING

One of the challenges of applying kernel learning algorithms to online scenarios is the linear growth of the kernel matrix. Given the kernel matrix $\boldsymbol{K}^{(t)} \in \mathbb{R}^{t \times t}$, the prototype model (Williams & Seeger,

2000) get the approximate kernel matrix $\hat{\boldsymbol{K}}^{(t)} = \boldsymbol{C}\boldsymbol{U}_{\text{fast}}\boldsymbol{C}^{\top}$ by solving the following problem:

$$\boldsymbol{U}_{\text{fast}} = \arg\min_{\boldsymbol{U}} ||\boldsymbol{C}\boldsymbol{U}\boldsymbol{C}^{\top} - \boldsymbol{K}^{(t)}||_F^2 = \boldsymbol{C}^{\dagger}\boldsymbol{K}^{(t)}(\boldsymbol{C}^{\top})^{\dagger}, \tag{1}$$

where $\boldsymbol{C}$ is the sketch and usually chooses the column-sampling matrix $\boldsymbol{S}_m \in \mathbb{R}^{t \times s_m}$ as the sketch matrix to reduce the size of the approximate kernel matrix, i.e., formulate $\boldsymbol{C} = \boldsymbol{K}^{(t)}\boldsymbol{S}_m \in \mathbb{R}^{t \times s_m}$.

However, it's essential to recognize that solving equation 1 can impose significant computational demands. Wang et al. (2016) proposed the sketched kernel matrix approximation problem as follows:

$$\boldsymbol{U}_{\text{fast}} = \arg\min_{\boldsymbol{U}} ||\boldsymbol{S}^{\top}\boldsymbol{C}\boldsymbol{U}\boldsymbol{C}^{\top}\boldsymbol{S} - \boldsymbol{S}^{\top}\boldsymbol{K}^{(t)}\boldsymbol{S}||_F^2 = (\boldsymbol{S}\boldsymbol{C})^{\dagger}\boldsymbol{S}^{\top}\boldsymbol{K}\boldsymbol{S}^{(t)}((\boldsymbol{S}\boldsymbol{C})^{\top})^{\dagger}, \tag{2}$$

where $\boldsymbol{S}$ can be different sketching matrices to reduce the complexity of equation 1. In this paper, we choose the randomized sketch matrix SJLT (defined in Appendix B), i.e., $\boldsymbol{S} = \boldsymbol{S_p} \in \mathbb{R}^{t \times s_p}$.

Therefore, we can maintain some small sketches for approximation instead of storing the entire kernel matrix. We denote an SJLT $\boldsymbol{S}_p^{(t+1)} \in \mathbb{R}^{(t+1) \times s_p}$ and a column-sampling matrix $\boldsymbol{S}_m^{(t+1)} \in \mathbb{R}^{(t+1) \times s_m}$. At round $t + 1$, a new example $\boldsymbol{x}_{t+1}$ arrives, and the kernel matrix $\boldsymbol{K}^{(t+1)} \in \mathbb{R}^{(t+1) \times (t+1)}$ can be approximated by $\hat{\boldsymbol{K}}^{(t+1)} = \boldsymbol{C}_m^{(t+1)}\boldsymbol{U}_{\text{fast}}\boldsymbol{C}_m^{(t+1)\top}$, where $\boldsymbol{C}_m^{(t+1)} = \boldsymbol{K}^{(t+1)}\boldsymbol{S}_m^{(t+1)} \in \mathbb{R}^{(t+1) \times s_m}$.

$\boldsymbol{U}_{\text{fast}}$ is derived by solving the sketched kernel matrix approximation problem, as in equation 2:

$$\boldsymbol{U}_{\text{fast}} = \left(\boldsymbol{\Phi}_{pm}^{(t+1)}\right)^{\dagger} \boldsymbol{\Phi}_{pp}^{(t+1)} \left(\boldsymbol{\Phi}_{pm}^{(t+1)\top}\right)^{\dagger} \in \mathbb{R}^{s_m \times s_m}, \tag{3}$$

where

$$\boldsymbol{\Phi}_{pm}^{(t+1)} = \boldsymbol{S}_p^{(t+1)\top}\boldsymbol{C}_m^{(t+1)} \in \mathbb{R}^{s_p \times s_m}, \quad \boldsymbol{\Phi}_{pp}^{(t+1)} = \boldsymbol{S}_p^{(t+1)\top}\boldsymbol{K}^{(t+1)}\boldsymbol{S}_p^{(t+1)} \in \mathbb{R}^{s_p \times s_p}. \tag{4}$$

Next, we construct the time-varying explicit feature mapping. For simplicity, we begin with rank-$k$ SVD. Since the elements of the kernel matrix are equal to the inner product of the corresponding points after feature mapping, i.e. $\boldsymbol{K}_{i,j} = \phi(x_i)\phi(x_j)^{\top}$. Once we build the approximate kernel matrix by equation 3, we can obtain a time-varying feature mapping through the rank-$k$ SVD. Specifically, if

$$\boldsymbol{\Phi}_{pp}^{(t+1)} \approx \boldsymbol{V}^{(t+1)}\boldsymbol{\Sigma}^{(t+1)}\boldsymbol{V}^{(t+1)\top} \in \mathbb{R}^{s_p \times s_p}, \tag{5}$$

where $\boldsymbol{V}^{(t+1)} \in \mathbb{R}^{s_p \times k}, \boldsymbol{\Sigma}^{(t+1)} \in \mathbb{R}^{k \times k}$ and rank $k \leq s_p$, we can update the time-varying explicit feature mapping at round $t + 1$ by

$$\boldsymbol{\phi}_{t+2}(\cdot) = ([\kappa(\cdot, \tilde{\boldsymbol{x}}_1), \kappa(\cdot, \tilde{\boldsymbol{x}}_2), ..., \kappa(\cdot, \tilde{\boldsymbol{x}}_{s_m})]\boldsymbol{Z}_{t+1})^{\top} \in \mathbb{R}^k,$$

where $\{\tilde{\boldsymbol{x}}_i\}_{i=1}^{s_m}$ are the sampled columns by $\boldsymbol{S}_m^{(t+1)}$, and $\boldsymbol{Z}_{t+1} = \left(\boldsymbol{\Phi}_{pm}^{(t+1)}\right)^{\dagger} \boldsymbol{V}^{(t+1)} \left(\boldsymbol{\Sigma}^{(t+1)}\right)^{\frac{1}{2}}$.

### 3.2 Novel Decomposition Method for Efficient Feature Mapping Updating

While it is possible to update the feature mapping by directly applying rank-$k$ SVD to $\boldsymbol{\Phi}_{pp} \in \mathbb{R}^{s_p \times s_p}$, this approach proves inefficient for online learning scenarios. Specifically, the standard rank-$k$ SVD incurs a time complexity of $O\left(s_p^3\right)$ at each update round, making it impractical for scenarios with a high volume of updates. To address these limitations, we propose **TISVD** (Truncated Incremental Singular Value Decomposition), a novel incremental SVD method explicitly tailored to decomposing sketches. TISVD offers linear time and space complexity concerning the sketch size $s_p$, efficiently addressing the computational challenges posed by frequent updates.

We will begin by presenting the construction of TISVD, which is well-suited for decomposing matrices with low-rank update properties. Without loss of generality, we denote a matrix at round $t$ as $\boldsymbol{M}^{(t)} = \boldsymbol{U}^{(t)}\boldsymbol{\Sigma}^{(t)}\boldsymbol{V}^{(t)\top}$. In the $(t + 1)$-th round, $\boldsymbol{M}^{(t)}$ is updated by low-rank matrices $\boldsymbol{A}, \boldsymbol{B} \in \mathbb{R}^{s_p \times c}$ of rank $r \leq c \ll s_p$:

$$\boldsymbol{M}^{(t+1)} = \boldsymbol{M}^{(t)} + \boldsymbol{A}\boldsymbol{B}^{\top} = \boldsymbol{U}^{(t+1)}\boldsymbol{\Sigma}^{(t+1)}\boldsymbol{V}^{(t+1)\top}. \tag{6}$$

Our objective is to directly update the singular matrices $\boldsymbol{U}^{(t)}, \boldsymbol{\Sigma}^{(t)}$ and $\boldsymbol{V}^{(t)}$ using low-rank update matrices $\boldsymbol{A}$ and $\boldsymbol{B}$, resulting in $\boldsymbol{U}^{(t+1)}, \boldsymbol{\Sigma}^{(t+1)}$ and $\boldsymbol{V}^{(t+1)}$. First, we formulate orthogonal matrices

through orthogonal projection and vertical projection. Let $\boldsymbol{P}, \boldsymbol{Q}$ denote orthogonal basis of the column space of $\left(\boldsymbol{I} - \boldsymbol{U}^{(t)}\boldsymbol{U}^{(t)\top}\right)\boldsymbol{A}$, $\left(\boldsymbol{I} - \boldsymbol{V}^{(t)}\boldsymbol{V}^{(t)\top}\right)\boldsymbol{B}$, respectively. We set $\boldsymbol{R}_A \doteq \boldsymbol{P}^{\top}\left(\boldsymbol{I} - \boldsymbol{U}^{(t)}\boldsymbol{U}^{(t)\top}\right)\boldsymbol{A}$ and $\boldsymbol{R}_B \doteq \boldsymbol{Q}^{\top}\left(\boldsymbol{I} - \boldsymbol{V}^{(t)}\boldsymbol{V}^{(t)\top}\right)\boldsymbol{B}$. Then, we can transform equation 6 into

$$\boldsymbol{M}^{(t+1)} = \begin{bmatrix} \boldsymbol{U}^{(t)} & \boldsymbol{P} \end{bmatrix} \boldsymbol{H} \begin{bmatrix} \boldsymbol{V}^{(t)} & \boldsymbol{Q} \end{bmatrix}^{\top}, \tag{7}$$

where

$$\boldsymbol{H} = \begin{bmatrix} \boldsymbol{\Sigma}^{(t)} & \boldsymbol{0} \\ \boldsymbol{0} & \boldsymbol{0} \end{bmatrix} + \begin{bmatrix} \boldsymbol{U}^{(t)\top}\boldsymbol{A} \\ \boldsymbol{R}_A \end{bmatrix} \begin{bmatrix} \boldsymbol{V}^{(t)\top}\boldsymbol{B} \\ \boldsymbol{R}_B \end{bmatrix}^{\top} \in \mathbb{R}^{(k+c)\times(k+c)}. \tag{8}$$

Subsequently, as the size of $\boldsymbol{H}$ is smaller than $\boldsymbol{M}^{(t+1)}$, an efficient computation of $\tilde{\boldsymbol{U}}_k, \tilde{\boldsymbol{V}}_k$, and $\tilde{\boldsymbol{\Sigma}}_k$ can be obtained by performing a truncated rank-$k$ SVD on $\boldsymbol{H}$. Since the matrices on the left and right sides are column orthogonal, we finally obtain $\boldsymbol{U}^{(t+1)}, \boldsymbol{V}^{(t+1)}$, and $\boldsymbol{\Sigma}^{(t+1)}$ at round $t+1$:

$$\boldsymbol{U}^{(t+1)} = \begin{bmatrix} \boldsymbol{U}^{(t)} & \boldsymbol{P} \end{bmatrix} \tilde{\boldsymbol{U}}_k, \quad \boldsymbol{V}^{(t+1)} = \begin{bmatrix} \boldsymbol{V}^{(t)} & \boldsymbol{Q} \end{bmatrix} \tilde{\boldsymbol{V}}_k, \quad \boldsymbol{\Sigma}^{(t+1)} = \tilde{\boldsymbol{\Sigma}}_k. \tag{9}$$

Then, we will elucidate how TISVD can be employed in the context of online kernel learning, leading to a substantial reduction in the computational overhead associated with updating feature mapping. Motivated by Zhang & Liao (2019), $\boldsymbol{\Phi}_{pp}^{(t+1)}$ can be updated by low-rank matrices, i.e., $\boldsymbol{\Phi}_{pp}^{(t+1)} = \boldsymbol{\Phi}_{pp}^{(t)} + \Delta_1 \Delta_2^{\top}$, where $\Delta_1, \Delta_2 \in \mathbb{R}^{s_p \times 3}$ (details in Appendix C). Building upon this foundation, we can employ TISVD to establish an efficient mechanism for the incremental maintenance of singular matrices. More precisely, we update $\boldsymbol{V}^{(t+1)}$ and $\boldsymbol{\Sigma}^{(t+1)}$ using their previous counterparts, $\boldsymbol{V}^{(t)}$ and $\boldsymbol{\Sigma}^{(t)}$, along with a low-rank update $\Delta_1, \Delta_2$, as in equation 9.

Compared to rank-$k$ SVD, TISVD yields significant improvements by reducing the time complexity from $O\left(s_p^3\right)$ to $O\left(s_p k + k^3\right)$ and the space complexity from $O\left(s_p^2\right)$ to $O\left(s_p k + k^2\right)$. TISVD efficiently constructs the feature mapping in linear time, eliminating the need to store the entire matrix. This renders it a practical decomposition scheme for OKL. The pseudocode of TISVD and further discussions are presented in the Appendix D, E due to space constraints.

## 3.3 APPLICATION TO SECOND-ORDER ONLINE KERNEL LEARNING

Since the efficient time-varying explicit feature mapping $\phi_t(\cdot)$ has been constructed, we can formulate the approximate hypothesis $f_t(\boldsymbol{x}_t)$ at round $t$ that is closed to the optimal hypothesis: $f_t(\boldsymbol{x}_t) = \boldsymbol{w}_t^{\top} \phi_t(\boldsymbol{x}_t)$, where $\boldsymbol{w}_t$ is the weight vector. On the basis of the hypothesis, we propose a two-stage online kernel learning procedure that follows the second-order update rules, named **FORKS** (Fast Second-Order Online Kernel Learning Using Incremental Sketching).

In the first stage, we simply collect the items with nonzero losses to the buffer $SV$ and perform the Kernelized Online Gradient Descent (KOGD) (Kivinen et al., 2004). When the size of the buffer reaches a fixed budget $B$, we calculate $\boldsymbol{K}_t$ and initialize sketch matrices $\boldsymbol{\Phi}_{pp}^{(t)}, \boldsymbol{\Phi}_{pm}^{(t)}$ in equation 3.

In the second stage, we adopt a periodic updating strategy for sketches. More precisely, we update $\boldsymbol{\Phi}_{pp}^{(t)}, \boldsymbol{\Phi}_{pm}^{(t)}$ by equation 11 (details in Appendix C) once for every $\rho$ examples, where $\rho \in [T - B]$ is defined as update cycle. Furthermore, we incrementally update the feature mapping $\phi_t(\cdot)$ by TISVD.

In addition to updating the feature mapping $\phi_t(\cdot)$, we perform second-order updates on the $\boldsymbol{w}_t$. Specifically, we update the hypothesis using Online Newton Step (ONS) (Hazan et al., 2007):

$$\boldsymbol{v}_{t+1} = \boldsymbol{w}_t - \boldsymbol{A}_t^{-1}\boldsymbol{g}_t, \quad \boldsymbol{w}_{t+1} = \boldsymbol{v}_{t+1} - \frac{h\left(\phi_{t+1}^{\top}\boldsymbol{v}_{t+1}\right)}{\phi_{t+1}^{\top}\boldsymbol{A}_t^{-1}\phi_{t+1}} \boldsymbol{A}_t^{-1}\phi_{t+1}, \tag{10}$$

where $\boldsymbol{g}_t = \nabla_{\boldsymbol{w}_t}\ell_t(\hat{y}_t)$ and $h(z) = \text{sign}(z)\max(|z| - C, 0)$. Moreover, for some parameters $\alpha > 0$ and $\sigma_i, \eta_i \geq 0$, we update $\boldsymbol{A}_t$ by $\boldsymbol{A}_t = \alpha \boldsymbol{I} + \sum_{i=0}^{t}(\sigma_i + \eta_i)\boldsymbol{g}_i\boldsymbol{g}_i^{\top}$. The second-order updates not only consider the gradient information but also utilize the curvature information of the loss function, leading to faster convergence rates.

At the start of a new update epoch, we incorporate a *reset* step before applying the gradient descent in the new embedded space. We update the feature mapping $\phi_t$ but reset $\boldsymbol{A}_t$ and $\boldsymbol{w}_t$. This step is taken to ensure that our starting point cannot be influenced by the adversary. By leveraging efficient

second-order updates, we can effectively converge to the optimal hypothesis within the current subspace. Furthermore, the reset of the descent procedure when transitioning between subspaces ensures a stable starting point and maintains a bounded regret throughout the entire process. Finally, we summarize the above stages into Algorithm 1.

---

**Algorithm 1:** FORKS

**Input:** Data stream $\{(\boldsymbol{x}_t, y_t)\}_{t=1}^T$, sketch size $s_p$, sample size $s_m$, rank $k$, budget $B$, update cycle $\rho$, regularizer $\alpha$, number of blocks $d$

**Output:** Predicted label $\{\hat{y}_t\}_{t=1}^T$

**for** $t \leftarrow 1, \ldots, T$ **do**
  Receive $\boldsymbol{x}_t$ and Predict $\hat{y}_t = \mathrm{sgn}\left(\boldsymbol{\phi}_t^\top \boldsymbol{w}_t\right)$
  **if** $|SV| < B$ **then**
    $SV_{t+1} \leftarrow SV_t \cup \{\boldsymbol{x}_t\}$ whenever the loss is nonzero
    Update hypothesis by KOGD
  **else**
    **if** $|SV| = B$ **then**
      Initialize $\boldsymbol{\Phi}_{pp}^{(t)}, \boldsymbol{\Phi}_{pm}^{(t)}$ as in equation 3, the mapping $\boldsymbol{\phi}_{t+1}$, and the weight $\boldsymbol{w}_{t+1}$
    **else if** $t \bmod \rho = 1$ **then**
      Update $\boldsymbol{\Phi}_{pp}^{(t)}, \boldsymbol{\Phi}_{pm}^{(t)}$ using rank-1 modifications
      Update $\boldsymbol{\phi}_{t+1}$ by TISVD
      $\boldsymbol{A}_t \leftarrow \alpha \boldsymbol{I}, \quad \boldsymbol{w}_t \leftarrow \boldsymbol{0}$
    **else**
      $\boldsymbol{\Phi}_{pp}^{(t)} \leftarrow \boldsymbol{\Phi}_{pp}^{(t-1)}, \quad \boldsymbol{\Phi}_{pm}^{(t)} \leftarrow \boldsymbol{\Phi}_{pm}^{(t-1)}, \quad \boldsymbol{\phi}_{t+1} \leftarrow \boldsymbol{\phi}_t$
    # Execute a second-order gradient descent
    Compute $\boldsymbol{g}_t \leftarrow \nabla_{\boldsymbol{w}_t} \ell_t(\hat{y}_t), \quad \boldsymbol{A}_{t+1} \leftarrow \boldsymbol{A}_t + (\sigma_i + \eta_i)\boldsymbol{g}_t \boldsymbol{g}_t^\top$
    Compute $\boldsymbol{v}_{t+1} \leftarrow \boldsymbol{w}_t - \boldsymbol{A}_t^{-1} \boldsymbol{g}_t, \quad \boldsymbol{w}_{t+1} \leftarrow \boldsymbol{v}_{t+1} - \frac{h\left(\boldsymbol{\phi}_{t+1}^\top \boldsymbol{v}_{t+1}\right)}{\boldsymbol{\phi}_{t+1}^\top \boldsymbol{A}_t^{-1} \boldsymbol{\phi}_{t+1}} \boldsymbol{A}_t^{-1} \boldsymbol{\phi}_{t+1}$

---

### 3.4 COMPLEXITY ANALYSIS OF FORKS

Given the budget $B$, our FORKS consists of three parts: (1) the first stage using KOGD, (2) the updating round in the second stage, and (3) the regular round in the second stage. At the first stage ($|SV| \leq B$), FORKS has constant time $O(B)$ and space complexities $O(B)$ per round.

The main computational complexity of FORKS during the update round stems from the matrix decomposition and inversion procedures. These processes are necessary for updating the feature mapping and performing second-order gradient updates, respectively. Our proposed TISVD reduce the time complexity of $\boldsymbol{\Phi}_{pp}$ decomposition from $O\left(s_p^3\right)$ to $O\left(s_p k + k^3\right)$, where $s_p$ is the sketch size of $\boldsymbol{S}_p$ and $k$ is the rank in TISVD. A naive implementation of the second-order update requires $O\left(k^3\right)$ per-step time and has a space complexity of $O\left(k^2\right)$ necessary to store the Hessian $\boldsymbol{A}_t$. However, by taking advantage of the fact that $\boldsymbol{A}_t$ is constructed using rank-1 modification, we can reduce the per-step cost to $O\left(k^2\right)$. We denote the update cycle as $\rho$ and $\mu = B + \left\lfloor \frac{T-B}{\rho} \right\rfloor$. To summarize, the time complexity of FORKS at each updating round is $O\left(\mu + s_p k^2 + s_m s_p k + k^3\right)$ and the space complexity of FORKS is $O\left(\mu + s_p k + s_m s_p + k^2\right)$, where $s_m$ is the sketch size of $\boldsymbol{S}_m$.

In online learning, the main time consumption of FORKS is the regular round. At each regular round, the time complexity of FORKS is $O\left(s_m k + k^2\right)$. Since we set $s_m < s_p < B$ in the experiments, our FORKS enjoys a time complexity of $O\left(Bk + k^2\right)$ per step, which is close to the first-order methods NOGD and SkeGD. The current state-of-the-art second-order online kernel learning method, PROS-N-KONS, presents a time complexity of $O\left(B^2\right)$ per step, making it less practical for large-scale online learning scenarios. In contrast, FORKS introduces substantial advancements by reducing the time complexity from $O\left(B^2\right)$ to $O\left(Bk + k^2\right)$, leading to more efficient computations.

# 4 REGRET ANALYSIS

In this section, we provide the regret analysis for the proposed second-order online kernel learning algorithm. We begin by making the following assumptions about the loss functions.

**Assumption 1** (Lipschitz Continuity). *$\ell$ is Lipschitz continuous with the Lipschitz constant $L_{\text{Lip}}$, i.e.,* $\|\nabla\ell(\boldsymbol{w})\|_2 \leq L_{\text{Lip}}$.

**Assumption 2** (Directional Curvature). *Let $L_{\text{Cur}} \geq 0$. Then, for any vectors $\boldsymbol{w}_1, \boldsymbol{w}_2$, the convex function $\ell$ satisfies the following condition: $\ell(\boldsymbol{w}_1) \geq \ell(\boldsymbol{w}_2) + \langle\nabla\ell(\boldsymbol{w}_2), \boldsymbol{w}_1 - \boldsymbol{w}_2\rangle + \frac{L_{\text{Cur}}}{2}\langle\nabla\ell(\boldsymbol{w}_2), \boldsymbol{w}_1 - \boldsymbol{w}_2\rangle^2$.*

In practical scenarios, the assumption of *strong convexity* may not always hold as it imposes constraints on the convexity of losses in all directions. A more feasible approach is to relax this assumption by demanding strong convexity only in the gradient direction, which is a weaker condition as indicated by the two assumptions above. For example, exp-concave losses like squared loss and squared hinge loss satisfy the condition in Assumption 2.

**Assumption 3** (Matrix Product Preserving). *Let $\boldsymbol{S}_p \in \mathbb{R}^{T \times s_p}$ be a sketch matrix, $\boldsymbol{U}_{\text{m}} \in \mathbb{R}^{T \times s_{\text{m}}}$ be a matrix with orthonormal columns, $\boldsymbol{U}_{\text{m}}^{\perp} \in \mathbb{R}^{T \times (T-s_{\text{m}})}$ be another matrix satisfying $\boldsymbol{U}_{\text{m}}\boldsymbol{U}_{\text{m}}^{\top} + \boldsymbol{U}_{\text{m}}^{\perp}(\boldsymbol{U}_{\text{m}}^{\perp})^{\top} = \boldsymbol{I}_T$ and $\boldsymbol{U}_{\text{m}}^{\top}\boldsymbol{U}_{\text{m}}^{\perp} = \boldsymbol{O}$, and $\delta_i$ ($i = 1, 2$) be the failure probabilities defined as follows:*

$$\Pr\left\{\|\boldsymbol{B}_i\boldsymbol{A}_i - \boldsymbol{B}_i\boldsymbol{S}_p\boldsymbol{S}_p^{\top}\boldsymbol{A}_i\|_F^2 > 2\|\boldsymbol{B}_i\boldsymbol{A}_i - \boldsymbol{B}_i\boldsymbol{S}_p\boldsymbol{S}_p^{\top}\boldsymbol{A}_i\|_F^2/(\delta_i s_p)\right\} \leq \delta_i, \quad i = 1, 2,$$

*where $\boldsymbol{A}_1 = \boldsymbol{U}_{\text{m}}$, $\boldsymbol{B}_1 = \boldsymbol{I}_T$, $\boldsymbol{A}_2 = \boldsymbol{U}_{\text{m}}^{\perp}(\boldsymbol{U}_{\text{m}}^{\perp})^{\top}\boldsymbol{K}$, $\boldsymbol{B}_2 = \boldsymbol{U}_{\text{m}}^{\top}$, $\boldsymbol{K} \in \mathbb{R}^{T \times T}$ is a kernel matrix.*

The conditions stated in Assumption 3 can be satisfied by several sketch matrices, such as SJLT (Woodruff, 2014). Given the loss $\ell_t(\boldsymbol{w}_t) := \ell_t(f_t) = \ell(f_t(\boldsymbol{x}_t), y_t), \forall t \in [T]$ satisfies the conditions in Assumption 1 and Assumption 2, we bound the following regret: $\text{Reg}_T(f^*) = \sum_{t=1}^{T}[\ell_t(\boldsymbol{w}_t) - \ell_t(f^*)]$, where $f^*$ denotes the optimal hypothesis in hindsight in the original reproducing kernel Hilbert space, i.e., $f^* = \arg\min_{f \in \mathcal{H}_\kappa} \sum_{t=1}^{T} \ell_t(f)$. Please note that although we optimize the objective function with the regularization term $\mathcal{L}_t(\boldsymbol{w}_t) = \ell_t(\boldsymbol{w}_t) + \lambda\|\boldsymbol{w}_t\|_2^2/2$, our focus is on the more fundamental unregularized regret and we provide its upper bound that is sublinear.

**Theorem 1** (Regret Bound of FORKS). *Let $\boldsymbol{K} \in \mathbb{R}^{T \times T}$ be a kernel matrix with $\kappa(\boldsymbol{x}_i, \boldsymbol{x}_j) \leq 1$, $\delta_0, \epsilon_0 \in (0, 1)$, and $k$ ($k \leq s_p$) be the rank in TISVD. Set the update cycle $\rho = \lfloor\theta(T - B)\rfloor$, $\theta \in (0, 1)$, and $d = \Theta(\log^3(s_m))$, in SJLT $\boldsymbol{S}_p \in \mathbb{R}^{T \times s_p}$. Assume the loss $\ell_t, \forall t \in [T]$ satisfies the conditions in Assumption 1 and Assumption 2, suppose that the parameters of updating $\boldsymbol{A}_t$ in FORKS satisfy $\eta_i = 0$ and $\sigma_i \geq L_{\text{Cur}} > 0$. Assume the eigenvalues of $\boldsymbol{K}$ decay polynomially with decay rate $\beta > 1$, and the SJLT $\boldsymbol{S}_p$ satisfies Assumption 3 with failure probabilities $\delta_1, \delta_2 \in (0, 1)$. If the sketch sizes of $\boldsymbol{S}_p$ and $\boldsymbol{S}_m$ satisfy*

$$s_p = \Omega\left(s_m \, \text{polylog}(s_m\delta_0^{-1})/\epsilon_0^2\right), \quad s_m = \Omega(C_{\text{Coh}}k\log k),$$

*where $C_{\text{Coh}}$ is the coherence of the intersection matrix of $\boldsymbol{K}$ which is constructed by $B + \lfloor(T - B)/\rho\rfloor$ examples independently of $T$, then with probability at least $1 - \delta$,*

$$\text{Reg}_T(f^*) \leq \frac{\alpha D_{\boldsymbol{w}}^2}{2} + \frac{k}{2L_{\text{Cur}}}O(\log T) + \frac{\lambda}{2}\|f^*\|_{\mathcal{H}_\kappa}^2 +$$

$$\frac{1}{\lambda(\beta - 1)}\left(\frac{3}{2} - \frac{B + \lfloor(T - B)/\rho\rfloor}{T}\right) + \frac{\sqrt{1 + \epsilon}}{\lambda}O(\sqrt{B}).$$

*where $\delta = \delta_0 + \delta_1 + \delta_2$, $D_{\boldsymbol{w}}$ is the diameter of the weight vector space of the hypothesis on the incremental sketches, $\epsilon$ is defined as:*

$$\sqrt{\epsilon} = 2\gamma\sqrt{T/(\delta_1\delta_2)} + \sqrt{2\gamma/\delta_2}\left(\epsilon_0^2 + 2\epsilon_0 + 2\right), \quad \gamma = s_m/s_p.$$

**Remark 1.** *In Theorem 1, the assumption of polynomial decay for eigenvalues of the kernel matrix is a widely applicable assumption, satisfied by shift-invariant kernels, finite rank kernels, and convolution kernels (Liu & Liao, 2015; Belkin, 2018). Setting the update cycle $\rho = \lfloor\theta(T - B)\rfloor$, $\theta \in (0, 1)$, and the sketch size ratio $\gamma = O(\log(T)/\sqrt{T})$, we can obtain the optimal regret upper bound of order $O(\log(T))$ for second-order online kernel learning (Hazan, 2016).*

**Remark 2.** *It's worth noting that when $L_{\mathrm{Cur}} = 0$, Assumption 2 essentially enforces convexity. In the worst case when $L_{\mathrm{Cur}} = 0$, the regret bound in the convex case degenerates to $O(\sqrt{T})$. The detailed proof is included in Appendix G.*

Table 1 provides a comprehensive comparison of the theoretical results of different budget online kernel learning algorithms. Analyzing the results reveals that the proposed FORKS algorithm achieves a tighter logarithmic regret bound compared to existing first-order online kernel learning algorithms, while significantly reducing the computational time required for second-order online optimization. Specifically, FORKS effectively reduces the time complexity of the existing second-order algorithm from quadratic w.r.t. the budget to linear, making it comparable to first-order algorithms.

Table 1: Comparison on different budget online kernel learning algorithms, where $B$ is the budget, $D$ is the feature dimension, and $k$ is the truncated rank in matrix decomposition.

| Algorithms | Optimization | Update Time | Regret Bound |
|---|---|---|---|
| RBP | First-Order | $O(B)$ | $O(\sqrt{T})$ |
| BOGD | First-Order | $O(B)$ | $O(\sqrt{T})$ |
| FOGD | First-Order | $O(D)$ | $O(\sqrt{T})$ |
| BPA-S | First-Order | $O(B)$ | $O(\sqrt{T})$ |
| Projectron | First-Order | $O(B^2)$ | $O(\sqrt{T})$ |
| NOGD | First-Order | $O(Bk)$ | $O(\sqrt{T})$ |
| SkeGD | First-Order | $O(Bk)$ | $O(\sqrt{T})$ |
| PROS-N-KONS | Second-Order | $O(B^2)$ | $O(\log T)$ |
| FORKS (Ours) | Second-Order | $O(Bk + k^2)$ | $O(\log T)$ |

## 5 EXPERIMENTS

In this section, we conduct experiments to evaluate the performance of FORKS on a wide variety of datasets. The details of datasets and experimental setup are presented in Appendix H, I.

### 5.1 EXPERIMENTS UNDER A FIXED BUDGET

In this section, we demonstrate the performance of FORKS under a fixed budget, employing six widely recognized classification benchmark datasets. We compare FORKS with the existing budgeted-based online learning algorithms, including RBP (Cavallanti et al., 2007), BPA-S (Wang & Vucetic, 2010), BOGD (Zhao et al., 2012), FOGD, NOGD (Lu et al., 2016a), Projectron (Orabona et al., 2008), PROS-N-KONS (Calandriello et al., 2017a), and SkeGD (Zhang & Liao, 2019). We implement the above models with the help of the LIBOL v0.3.0 toolbox toolbox [1]. All algorithms are trained using hinge loss, and their performance is measured by the average online mistake rate.

For all the algorithms, we set a fixed budget $B = 50$ for small datasets ($N \leq 10000$) and $B = 100$ for large datasets. Furthermore, we set buffer size $\tilde{B} = 2B, \gamma = 0.2, s_p = B, s_m = \gamma s_p, \theta = 0.3$, and update cycle $\rho = \lfloor \theta N \rfloor$ in SkeGD and FORKS if not specially specified. For algorithms with rank-$k$ approximation, we uniformly set $k = 0.1B$. Besides, we use the same experimental settings for FOGD (feature dimension = $4B$). The results are presented in Table 2. Our FORKS shows the best performance on most datasets and the suboptimal performance on german and ijcnn1. The update time of FORKS is comparable to that of the majority of first-order algorithms, including NOGD and SkeGD. Besides, FORKS is significantly more efficient than the existing second-order method PROS-N-KONS in large-scale datasets such as codrna and w7a.

Then, we conduct experiments to evaluate how TISVD affects the performance of the algorithm. We use the same experimental setup in codrna and vary the update rate $\theta$ from 0.5 to 0.0005. Figure 1 demonstrates that TISVD maintains efficient decomposition speed without excessively reducing performance. Furthermore, considering that frequent updates can potentially result in an elevated loss, it is essential to carefully choose an optimal update cycle that strikes a balance between achieving superior accuracy and maintaining efficiency.

---

[1]http://libol.stevenhoi.org/

Table 2: Comparisons among RBP, BPA-S, BOGD, Projectron, NOGD, PROS-N-KONS, SkeGD, FOGD and our FORKS w.r.t. the mistake rates (%) and the running time (s). The best result is highlighted in **bold** font, and the second best result is underlined.

| Algorithm | german | | svmguide3 | | spambase | |
|---|---|---|---|---|---|---|
| | Mistake rate | Time | Mistake rate | Time | Mistake rate | Time |
| RBP | $38.830 \pm 0.152$ | 0.003 | $29.698 \pm 1.644$ | 0.003 | $35.461 \pm 0.842$ | 0.025 |
| BPA-S | $35.235 \pm 0.944$ | 0.004 | $29.027 \pm 0.732$ | 0.004 | $34.394 \pm 2.545$ | 0.039 |
| Projectron | $36.875 \pm 1.403$ | 0.003 | $25.060 \pm 0.373$ | 0.003 | $32.659 \pm 0.914$ | 0.031 |
| BOGD | $33.705 \pm 1.446$ | 0.007 | $29.904 \pm 1.653$ | 0.006 | $32.859 \pm 0.478$ | 0.049 |
| FOGD | $30.915 \pm 0.845$ | 0.025 | $30.024 \pm 0.787$ | 0.022 | $\mathbf{25.651 \pm 0.349}$ | 0.175 |
| NOGD | $26.715 \pm 0.552$ | 0.014 | $\underline{19.964 \pm 0.077}$ | 0.008 | $31.003 \pm 0.751$ | 0.077 |
| PROS-N-KONS | $31.235 \pm 0.939$ | 1.017 | $24.529 \pm 0.561$ | 0.015 | $32.227 \pm 0.678$ | 6.638 |
| SkeGD | $\mathbf{25.170 \pm 0.391}$ | 0.009 | $19.976 \pm 0.105$ | 0.007 | $32.413 \pm 1.886$ | 0.067 |
| FORKS | $\underline{26.425 \pm 0.562}$ | 0.008 | $\mathbf{19.710 \pm 0.557}$ | 0.009 | $\underline{30.662 \pm 0.670}$ | 0.070 |

| Algorithm | codrna | | w7a | | ijcnn1 | |
|---|---|---|---|---|---|---|
| | Mistake rate | Time | Mistake rate | Time | Mistake rate | Time |
| RBP | $22.644 \pm 0.262$ | 0.210 | $5.963 \pm 0.722$ | 0.945 | $21.024 \pm 0.578$ | 0.633 |
| BPA-S | $17.029 \pm 0.303$ | 0.313 | $3.001 \pm 0.045$ | 1.145 | $11.114 \pm 0.064$ | 0.747 |
| Projectron | $19.257 \pm 4.688$ | 0.341 | $3.174 \pm 0.014$ | 0.965 | $9.478 \pm 0.001$ | 0.621 |
| BOGD | $17.305 \pm 0.146$ | 0.507 | $3.548 \pm 0.164$ | 0.970 | $11.559 \pm 0.174$ | 0.724 |
| FOGD | $\underline{13.103 \pm 0.105}$ | 1.480 | $2.893 \pm 0.053$ | 2.548 | $9.674 \pm 0.105$ | 3.125 |
| NOGD | $17.915 \pm 3.315$ | 0.869 | $\underline{2.579 \pm 0.007}$ | 2.004 | $\mathbf{9.379 \pm 0.001}$ | 1.457 |
| PROS-N-KONS | $13.387 \pm 0.289$ | 114.983 | $3.016 \pm 0.007$ | 92.377 | $9.455 \pm 0.001$ | 5.000 |
| SkeGD | $13.274 \pm 0.262$ | 0.779 | $2.706 \pm 0.335$ | 2.093 | $11.898 \pm 1.440$ | 2.216 |
| FORKS | $\mathbf{12.795 \pm 0.360}$ | 0.918 | $\mathbf{2.561 \pm 0.038}$ | 2.240 | $\underline{9.381 \pm 0.001}$ | 2.480 |

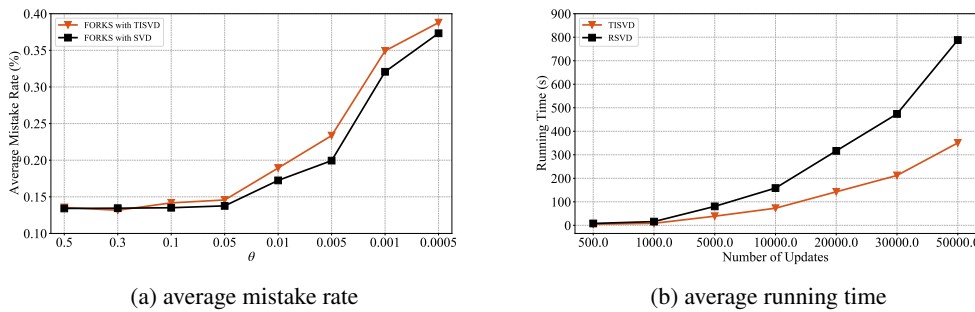

(a) average mistake rate          (b) average running time

Figure 1: The average mistake rates and average running time w.r.t. TISVD on codrna.

## 5.2 EXPERIMENTS UNDER ADVERSARIAL ENVIRONMENT

To empirically validate the algorithms under an adversarial environment, we build adversarial datasets using the benchmark codrna and german. We compare FORKS with first-order algorithms BOGD (Zhao et al., 2012), SkeGD (Zhang & Liao, 2019), NOGD (Lu et al., 2016a) and second-order algorithm PROS-N-KONS (Calandriello et al., 2017a) under the same budget $B = 200$. Besides, we set $\gamma = 0.2$, $s_p = 0.75B$, $s_m = \gamma s_p$, $k = 0.1B$ and update cycle $\rho = \lfloor 0.005(N - B) \rfloor$ in SkeGD and FORKS. Inspired by the adversarial settings in (Calandriello et al., 2017a; Zhang & Liao, 2019; Wang et al., 2018), we generate an online learning game with $b$ blocks. At each block, we extract an instance from the dataset and repeat it for $r$ rounds. In addition, the labels are flipped in each even block by multiplying them with -1. We set $b = 500$, $r = 10$ for codrna-1 and german-1.

Experimental results are presented in Table 3. It is observed that in the adversarial environment, the performance of all methods significantly decreases with the increase of adversarial changes except for FORKS. This is due to the fact that FORKS accurately captures the concept drifting through the incremental update of the sketch matrix and the execution of rapid second-order gradient descent. Moreover, FORKS maintains its computational efficiency comparable to first-order algorithms, thereby ensuring that improved performance is achieved without sacrificing computational time.

Table 3: Comparisons among BOGD, NOGD, PROS-N-KONS, SkeGD and our FORKS w.r.t. the mistake rates (%) and the running time (s). The best result is highlighted in **bold** font.

| Algorithm | codrna-1 | | german-1 | |
|---|---|---|---|---|
| | Mistake rate | Time | Mistake rate | Time |
| BOGD | $26.066 \pm 1.435$ | 0.029 | $32.131 \pm 1.079$ | 0.042 |
| NOGD | $29.780 \pm 1.257$ | 0.024 | $28.103 \pm 1.247$ | 0.040 |
| PROS-N-KONS | $21.299 \pm 1.364$ | 3.323 | $17.174 \pm 1.437$ | 0.477 |
| SkeGD | $24.649 \pm 5.087$ | 0.269 | $11.026 \pm 4.018$ | 0.113 |
| FORKS | $\mathbf{6.752 \pm 1.647}$ | 0.023 | $\mathbf{5.142 \pm 0.215}$ | 0.035 |

### 5.3 EXPERIMENTS ON LARGE-SCALE REAL-WORLD DATASETS

In this experiment, we evaluate the efficiency and effectiveness of FORKS on large-scale online learning tasks. We use KuaiRec, which is a real-world dataset collected from the recommendation logs of the video-sharing mobile app Kuaishou (Gao et al., 2022). We conduct experiments on the dense matrix of KuaiRec, which consists of $4,494,578$ instances with associated timestamps, making it an ideal benchmark for evaluating large-scale online learning tasks. We test the performance of the algorithm used in Section 5.3 under different budgets $B$ ranging from 100 to 500. To avoid excessive training time, we use a budgeted version of PROS-N-KONS that stops updating the dictionary at a maximum budget of $B_{\max} = 100$. Since the buffer size of PROS-N-KONS is data-dependent, we repeat the training process 20 times to compute the average error rate and the average time for comparison. In addition to the hinge loss, we use squared hinge loss to evaluate the performance of algorithms under the directional curvature conditions.

Figure 2 (a) shows the tradeoff between running time and the average mistake rate in the experiment using hinge loss. Figure 2 (b) shows the tradeoff between running time and the average mistake rate in the experiment using squared hinge loss. We observe that FORKS consistently achieves superior learning performance while maintaining comparable time costs to the other first-order algorithms, regardless of the loss function's shape. In particular, for squared hinge loss, both PROS-N-KONS and FORKS significantly outperform first-order models, highlighting the advantages of second-order methods under exp-concave losses. Additionally, we note that FORKS exhibits significantly higher efficiency compared to the second-order algorithm PROS-N-KONS. In fact, to achieve a similar online error rate, FORKS speeds up the running time by a factor of 3.

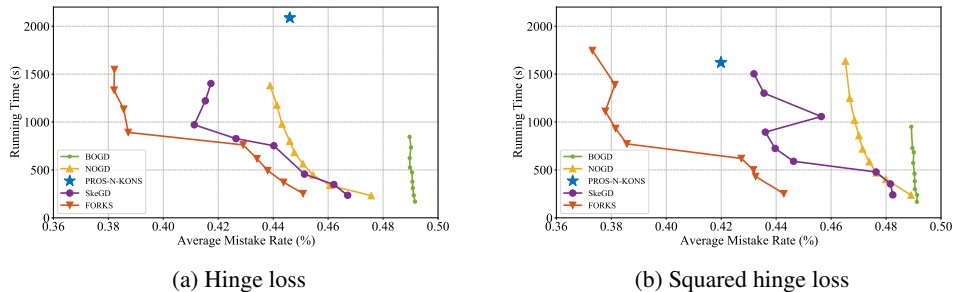

(a) Hinge loss                                      (b) Squared hinge loss

Figure 2: The tradeoff between running time and the average mistake rate on KuaiRec. As PROS-N-KONS utilizes an adaptive budget, it cannot modify computational costs, thereby being depicted as a single point in the figures.

## 6 CONCLUSION

This paper introduces FORKS, a fast second-order online kernel learning approach. FORKS leverages incremental sketching techniques to efficiently handle complex computations and incremental updates of data matrices and hypotheses, effectively addresses the challenge of concept drifting in data streams, and achieves a logarithmic regret bound, while maintaining linear time complexity with respect to the budget. Extensive experiments are conducted on real-world datasets to validate the superior scalability and robustness of FORKS, showcasing its potential for real-world online learning tasks.

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

## A    CODE FOR REPRODUCIBILITY

Code without datasets is provided in the supplementary material.

## B    MATRIX SKETCHING

Without loss of generality, given a matrix $M \in \mathbb{R}^{a \times b}$, the sketch of $M$ is defined as $MS \in \mathbb{R}^{a \times s}$, where $S \in \mathbb{R}^{b \times s}$ is a sketch matrix. In this paper, we introduce the Sparse Johnson-Lindenstrauss Transform and Column-sampling matrix as the sketch matrix (Charikar et al., 2002; Kane & Nelson, 2014).

**Sparse Johnson-Lindenstrauss Transform (SJLT)**: Randomized sketches via hash functions can be described in general using hash-based sketch matrices. We denote $\{h_k : \{1, ..., b\} \to \{1, ..., s_p\}\}$ and $\{g_k : \{1, ..., b\} \to \{-1/\sqrt{d}, 1/\sqrt{d}\}\}$ as two different $O(\log T)$-wise independent hash function sets, where $k \in \{1, ..., d\}$ and $d$ is the number of blocks. We denote SJLT by:

$$S = [S_1, ..., S_d] \in \mathbb{R}^{b \times s_p}$$

where $[S_k]_{i,j} = g(i)$ for $j = h_k(i)$ and $[S_k]_{i,j} = 0$ for $j \neq h_k(i)$.

**Column-sampling matrix**: We denote the Column-sampling matrix by $S_m \in \mathbb{R}^{b \times s_m}$, the columns of $S_m$ is obtained by uniformly sampling column vectors of $I \in \mathbb{R}^{b \times b}$.

## C    INCREMENTAL MAINTENANCE OF RANDOMIZED SKETCH

At round $t + 1$, a new example $x_{t+1}$ arrives, and the kernel matrix $K^{(t+1)} \in \mathbb{R}^{(t+1) \times (t+1)}$ can be represented as a bordered matrix and approximated using several small sketches as follows:

$$K^{(t+1)} = \begin{bmatrix} K^{(t)} & \psi^{(t+1)} \\ \psi^{(t+1)\top} & \kappa(x_{t+1}, x_{t+1}) \end{bmatrix} \approx C_m^{(t+1)} \left( \Phi_{pm}^{(t+1)} \right)^\dagger \Phi_{pp}^{(t+1)} \left( \Phi_{pm}^{(t+1)\top} \right)^\dagger C_m^{(t+1)\top},$$

where $\psi^{(t+1)} = [\kappa(x_{t+1}, x_1), \kappa(x_{t+1}, x_2), \ldots, \kappa(x_{t+1}, x_t)]^\top$.

The sketches can be represented as

$$\Phi_{pm}^{(t+1)} = S_p^{(t+1)\top} C_m^{(t+1)}, \; \Phi_{pp}^{(t+1)} = S_p^{(t+1)\top} C_p^{(t+1)},$$

where $C_m^{(t+1)} = K^{(t+1)} S_m^{(t+1)}, C_p^{(t+1)} = K^{(t+1)} S_p^{(t+1)}$.

The sketches are obtained using an SJLT $S_p^{(t+1)} \in \mathbb{R}^{(t+1) \times s_p}$ and a column-sampling matrix $S_m^{(t+1)} \in \mathbb{R}^{(t+1) \times s_m}$. We partition the sketch matrices into block matrices as $S_p^{(t+1)} = \left[ S_p^{(t)\top}, s_p^{(t+1)} \right]^\top, S_m^{(t+1)} = \left[ S_m^{(t)\top}, s_m^{(t+1)} \right]^\top$, where $s_m^{(t+1)} \in \mathbb{R}^{s_m}$ is a sub-sampling vector and $s_p^{(t+1)} \in \mathbb{R}^{s_p}$ is a new row vector of $S_p^{(t+1)}$ sharing the same hash functions.

Furthermore, we can update the sketches $\Phi_{pm}^{(t+1)}$ and $\Phi_{pp}^{(t+1)}$ using rank-1 modifications as follows:

1. Sketch $\boldsymbol{\Phi}_{pm}^{(t+1)}$

   The sketch $\boldsymbol{\Phi}_{pm}^{(t+1)}$ can be maintained as

$$
\begin{aligned}
&\boldsymbol{\Phi}_{pm}^{(t+1)} \\
&= \boldsymbol{S}_{\mathrm{p}}^{(t+1)\mathsf{T}} \boldsymbol{C}_{\mathrm{m}}^{(t+1)} \\
&= \boldsymbol{S}_{\mathrm{p}}^{(t+1)\mathsf{T}} \boldsymbol{K}^{(t+1)} \boldsymbol{S}_{\mathrm{m}}^{(t+1)} \\
&= [\boldsymbol{S}_{\mathrm{p}}^{(t)\mathsf{T}}, \boldsymbol{s}_{\mathrm{p}}^{(t+1)}]
\begin{bmatrix}
\boldsymbol{K}^{(t)} & \boldsymbol{\psi}^{(t+1)} \\
\boldsymbol{\psi}^{(t+1)\mathsf{T}} & \kappa(\boldsymbol{x}_{t+1}, \boldsymbol{x}_{t+1})
\end{bmatrix}
\begin{bmatrix}
\boldsymbol{S}_{\mathrm{m}}^{(t)} \\
\boldsymbol{s}_{\mathrm{m}}^{(t+1)\mathsf{T}}
\end{bmatrix} \\
&= \begin{bmatrix}
\boldsymbol{S}_{\mathrm{p}}^{(t)\mathsf{T}} \boldsymbol{K}^{(t)} + \boldsymbol{s}_{\mathrm{p}}^{(t+1)} \boldsymbol{\psi}^{(t+1)\mathsf{T}} \\
\boldsymbol{S}_{\mathrm{p}}^{(t)\mathsf{T}} \boldsymbol{\psi}^{(t+1)} + \kappa(\boldsymbol{x}_{t+1}, \boldsymbol{x}_{t+1}) \boldsymbol{s}_{\mathrm{p}}^{(t+1)}
\end{bmatrix}^{\mathsf{T}}
\begin{bmatrix}
\boldsymbol{S}_{\mathrm{m}}^{(t)} \\
\boldsymbol{s}_{\mathrm{m}}^{(t+1)\mathsf{T}}
\end{bmatrix} \\
&= \boldsymbol{S}_{\mathrm{p}}^{(t)\mathsf{T}} \boldsymbol{K}^{(t)} \boldsymbol{S}_{\mathrm{m}}^{(t)} + \boldsymbol{R}_{pm}^{(t+1)} + \boldsymbol{R}_{mp}^{(t+1)\mathsf{T}} + \boldsymbol{T}_{pm}^{(t+1)} \\
&= \boldsymbol{\Phi}_{pm}^{(t)} + \boldsymbol{R}_{pm}^{(t+1)} + \boldsymbol{R}_{mp}^{(t+1)\mathsf{T}} + \boldsymbol{T}_{pm}^{(t+1)},
\end{aligned}
$$

   where the modifications are performed using the following three rank-1 matrices

$$
\begin{aligned}
\boldsymbol{R}_{pm}^{(t+1)} &= \boldsymbol{s}_{\mathrm{p}}^{(t+1)} \boldsymbol{\psi}^{(t+1)\mathsf{T}} \boldsymbol{S}_{\mathrm{m}}^{(t)}, \\
\boldsymbol{R}_{mp}^{(t+1)} &= \boldsymbol{s}_{\mathrm{m}}^{(t+1)} \boldsymbol{\psi}^{(t+1)\mathsf{T}} \boldsymbol{S}_{\mathrm{p}}^{(t)}, \\
\boldsymbol{T}_{pm}^{(t+1)} &= \kappa(\boldsymbol{x}_{t+1}, \boldsymbol{x}_{t+1}) \boldsymbol{s}_{\mathrm{p}}^{(t+1)} \boldsymbol{s}_{\mathrm{m}}^{(t+1)\mathsf{T}}.
\end{aligned}
$$

2. Sketch $\boldsymbol{\Phi}_{pp}^{(t+1)}$

   For sketch $\boldsymbol{\Phi}_{pp}^{(t+1)}$, we have

$$
\begin{aligned}
&\boldsymbol{\Phi}_{pp}^{(t+1)} \\
&= \boldsymbol{S}_{\mathrm{p}}^{(t+1)\mathsf{T}} \boldsymbol{C}_{\mathrm{p}}^{(t+1)} \\
&= \boldsymbol{S}_{\mathrm{p}}^{(t+1)\mathsf{T}} \boldsymbol{K}^{(t+1)} \boldsymbol{S}_{\mathrm{p}}^{(t+1)} \\
&= [\boldsymbol{S}_{\mathrm{p}}^{(t)\mathsf{T}}, \boldsymbol{s}_{\mathrm{p}}^{(t+1)}]
\begin{bmatrix}
\boldsymbol{K}^{(t)} & \boldsymbol{\psi}^{(t+1)} \\
\boldsymbol{\psi}^{(t+1)\mathsf{T}} & \kappa(\boldsymbol{x}_{t+1}, \boldsymbol{x}_{t+1})
\end{bmatrix}
\begin{bmatrix}
\boldsymbol{S}_{\mathrm{p}}^{(t)} \\
\boldsymbol{s}_{\mathrm{p}}^{(t+1)\mathsf{T}}
\end{bmatrix} \\
&= \begin{bmatrix}
\boldsymbol{S}_{\mathrm{p}}^{(t)\mathsf{T}} \boldsymbol{K}^{(t)} + \boldsymbol{s}_{\mathrm{p}}^{(t+1)} \boldsymbol{\psi}^{(t+1)\mathsf{T}} \\
\boldsymbol{S}_{\mathrm{p}}^{(t)\mathsf{T}} \boldsymbol{\psi}^{(t+1)} + \kappa(\boldsymbol{x}_{t+1}, \boldsymbol{x}_{t+1}) \boldsymbol{s}_{\mathrm{p}}^{(t+1)}
\end{bmatrix}^{\mathsf{T}}
\begin{bmatrix}
\boldsymbol{S}_{\mathrm{p}}^{(t)} \\
\boldsymbol{s}_{\mathrm{p}}^{(t+1)\mathsf{T}}
\end{bmatrix} \\
&= \boldsymbol{S}_{\mathrm{p}}^{(t)\mathsf{T}} \boldsymbol{K}^{(t)} \boldsymbol{S}_{\mathrm{p}}^{(t)} + \boldsymbol{R}_{pp}^{(t+1)} + \boldsymbol{R}_{pp}^{(t+1)\mathsf{T}} + \boldsymbol{T}_{pp}^{(t+1)}, \\
&= \boldsymbol{\Phi}_{pp}^{(t)} + \boldsymbol{R}_{pp}^{(t+1)} + \boldsymbol{R}_{pp}^{(t+1)\mathsf{T}} + \boldsymbol{T}_{pp}^{(t+1)},
\end{aligned}
$$

   where the modifications are done by the following two rank-1 matrices

$$
\begin{aligned}
\boldsymbol{R}_{pp}^{(t+1)} &= \boldsymbol{s}_{\mathrm{p}}^{(t+1)} \boldsymbol{\psi}^{(t+1)\mathsf{T}} \boldsymbol{S}_{\mathrm{p}}^{(t)}, \\
\boldsymbol{T}_{pp}^{(t+1)} &= \kappa(\boldsymbol{x}_{t+1}, \boldsymbol{x}_{t+1}) \boldsymbol{s}_{\mathrm{p}}^{(t+1)} \boldsymbol{s}_{\mathrm{p}}^{(t+1)\mathsf{T}}.
\end{aligned}
$$

In summary, sketches can be updated through low-rank matrices:

$$
\begin{aligned}
\boldsymbol{\Phi}_{pm}^{(t+1)} &= \boldsymbol{\Phi}_{pm}^{(t)} + \boldsymbol{s}_{p}^{(t+1)} \boldsymbol{\psi}^{(t+1)\top} \boldsymbol{S}_{m}^{(t)} + \boldsymbol{S}_{p}^{(t)\top} \boldsymbol{\psi}^{(t+1)} \boldsymbol{s}_{m}^{(t+1)\top} + \kappa(\boldsymbol{x}_{t+1}, \boldsymbol{x}_{t+1}) \boldsymbol{s}_{p}^{(t+1)} \boldsymbol{s}_{m}^{(t+1)\top}, \\
\boldsymbol{\Phi}_{pp}^{(t+1)} &= \boldsymbol{\Phi}_{pp}^{(t)} + \boldsymbol{s}_{p}^{(t+1)} \boldsymbol{\psi}^{(t+1)\top} \boldsymbol{S}_{p}^{(t)} + \boldsymbol{S}_{p}^{(t)\top} \boldsymbol{\psi}^{(t+1)} \boldsymbol{s}_{p}^{(t+1)\top} + \kappa(\boldsymbol{x}_{t+1}, \boldsymbol{x}_{t+1}) \boldsymbol{s}_{p}^{(t+1)} \boldsymbol{s}_{p}^{(t+1)\top},
\end{aligned}
\tag{11}
$$

Specifically, the proposed TISVD method efficiently constructs the time-varying explicit feature mapping $\phi_t(\cdot)$ in equation 5 by setting $\boldsymbol{M} = \boldsymbol{\Phi}_{pp}$ and

$$
\boldsymbol{A} = \left[ \boldsymbol{s}_{p}^{(t+1)}, \boldsymbol{S}_{p}^{(t)\top} \boldsymbol{\psi}^{(t+1)}, \boldsymbol{s}_{p}^{(t+1)} \right], \quad \boldsymbol{B} = \left[ \boldsymbol{S}_{p}^{(t)\top} \boldsymbol{\psi}^{(t+1)}, \boldsymbol{s}_{p}^{(t+1)}, \kappa(\boldsymbol{x}_{t+1}, \boldsymbol{x}_{t+1}) \boldsymbol{s}_{p}^{(t+1)} \right]. \tag{12}
$$

# D  MORE DISCUSSION ABOUT TISVD

Current incremental SVD methods necessitate the prerequisite that the decomposition matrix adheres to a low-rank structure Brand (2006). When this low-rank condition isn't met, these methods devolve into traditional SVD. However, in online learning scenarios, the assurance of a low-rank decomposed sketch matrix isn't guaranteed. In this context, TISVD innovatively accomplishes incremental maintenance of singular value matrices without relying on low-rank assumptions, rendering it adapt to online learning algorithms founded on incremental sketching methodologies.

More precisely, given the matrix $\boldsymbol{A} = \boldsymbol{U}\boldsymbol{\Sigma}\boldsymbol{V}^{\top} \in \mathbb{R}^{n \times n}$, the conventional incremental SVD (ISVD) streamlines the process by omitting the rotation and re-orthogonalization of $\boldsymbol{U}$ and $\boldsymbol{V}$, leading to a time complexity of $O(nr + r^3)$. where $r$ denotes the matrix rank. Consequently, ISVD relies on the assumption that $r \ll n$ in order to effectively establish a linear-time SVD algorithm.

Nevertheless, in online learning scenarios, the sketch matrix earmarked for decomposition frequently fails to adhere to the low-rank characteristic, thereby rendering the direct application of ISVD ineffective in achieving linear time complexity. To counter this predicament, we have integrated truncation techniques within the framework of traditional incremental SVD methods. This adaptation yields a time complexity of $O(nr_t + r_t^3)$, with $r_t$ signifying the predetermined truncated rank. Crucially, this truncation innovation positions TISVD as a linear incremental SVD technique that stands independent of low-rank assumptions.

As previously discussed, applying ISVD directly to online learning algorithms isn't viable. Recognizing the substantial enhancement that accelerated decomposition feature mapping can offer to the performance of online kernel learning algorithms, prevailing research employs the randomized SVD algorithm to expedite these algorithms Wan & Zhang (2021); Zhang & Liao (2019). However, it's important to highlight that, unlike the incremental SVD method, the randomized SVD is a rapid SVD technique reliant on random matrices and lacks the capability to perform incremental updates on singular value matrices. Furthermore, it's worth noting that the time complexity of randomized SVD is $O(n^2r + r^3)$, which is comparatively slower than TISVD's $O(nr + r^3)$.

We have compared TISVD with the rank-$k$ truncated SVD in section 5.2. We further construct an experiment to test the performance of randomized SVD. We initialize a random Gaussian matrix $\boldsymbol{A} \in \mathbb{R}^{100 \times 100}$ and use random Gaussian matrix $B, C \in \mathbb{R}^{m \times 100}$ as low rank update. We set $m = 3, k = 30$ and update $\boldsymbol{A}$ 500 to 50000 times respectively.

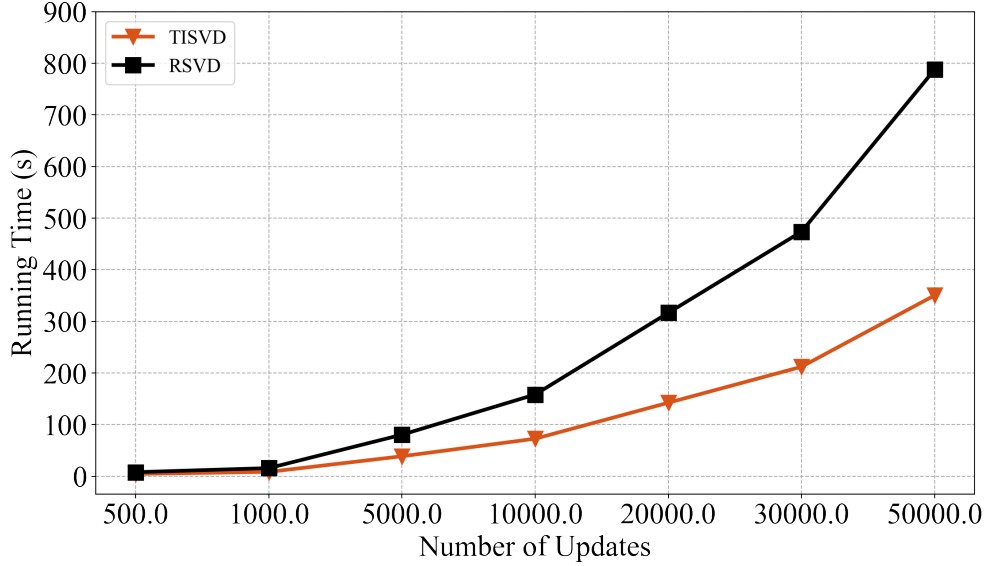

Figure 3: The comparison of running time between TISVD and RSVD

From Figure 3, we see that TISVD continues to show desirable decomposition performance. Both TIVD and randomized SVD can reduce the size of the decomposed matrix through the low-rank approximation matrix, thereby accelerating the algorithm. However, randomized SVD enjoys a time complexity of $O(n^2 k + k^3)$, which is worse than $O(nk + k^3)$. Besides, TISVD uses incremental updates to update the singular value matrix, which is more scalable for online learning algorithms based on incremental sketching.

## E   THE PSEUDO CODE OF TISVD

---

**Algorithm 2: TISVD**

---

**Input:** Rank-$k$ singular matrix $\boldsymbol{U}^{(t)}$, $\boldsymbol{V}^{(t)}$ and $\boldsymbol{\Sigma}^{(t)}$ at round $t$, low-rank matrix $\boldsymbol{A}$ and $\boldsymbol{B}$, truncated rank $k$

**Output:** Rank-$k$ singular matrix $\boldsymbol{U}^{(t+1)}$, $\boldsymbol{V}^{(t+1)}$ and $\boldsymbol{\Sigma}^{(t+1)}$ at round $t+1$

$\boldsymbol{U_A} \leftarrow \left( \boldsymbol{I} - \boldsymbol{U}^{(t)} \boldsymbol{U}^{(t)\top} \right) \boldsymbol{A}, \boldsymbol{V_B} \leftarrow \left( \boldsymbol{I} - \boldsymbol{V}^{(t)} \boldsymbol{V}^{(t)\top} \right) \boldsymbol{B}$

Compute orthogonal basis $\boldsymbol{P}, \boldsymbol{Q}$ of the column space of $\boldsymbol{U_A}, \boldsymbol{V_B}$, respectively.

$\boldsymbol{R_A} \leftarrow \boldsymbol{P}^\top \left( \boldsymbol{I} - \boldsymbol{U}^{(t)} \boldsymbol{U}^{(t)\top} \right) \boldsymbol{A}, \boldsymbol{R_B} \leftarrow \boldsymbol{Q}^\top \left( \boldsymbol{I} - \boldsymbol{V}^{(t)} \boldsymbol{V}^{(t)\top} \right) \boldsymbol{B}$

$\boldsymbol{H} \leftarrow \begin{bmatrix} \boldsymbol{\Sigma}^{(t)} & \boldsymbol{0} \\ \boldsymbol{0} & \boldsymbol{0} \end{bmatrix} + \begin{bmatrix} \boldsymbol{U}^{(t)\top} \boldsymbol{A} \\ \boldsymbol{R_A} \end{bmatrix} \begin{bmatrix} \boldsymbol{V}^{(t)\top} \boldsymbol{B} \\ \boldsymbol{R_B} \end{bmatrix}^\top$

Compute $\tilde{\boldsymbol{U}}_k, \tilde{\boldsymbol{V}}_k$ and $\tilde{\boldsymbol{\Sigma}}_k$ from rank-$k$ SVD of $\boldsymbol{H}$

# Update singular matrix.

$\boldsymbol{U}^{(t+1)} \leftarrow \begin{bmatrix} \boldsymbol{U}^{(t)} & \boldsymbol{P} \end{bmatrix} \tilde{\boldsymbol{U}}$

$\boldsymbol{V}^{(t+1)} \leftarrow \begin{bmatrix} \boldsymbol{V}^{(t)} & \boldsymbol{Q} \end{bmatrix} \tilde{\boldsymbol{V}}$

$\boldsymbol{\Sigma}^{(t+1)} \leftarrow \tilde{\boldsymbol{\Sigma}}$

**return** $\boldsymbol{U}^{(t+1)}, \boldsymbol{V}^{(t+1)}, \boldsymbol{\Sigma}^{(t+1)}$

---

## F   PROOF OF THEOREM 1

*Proof.* We can refine the representation of the difference in losses between $f^*$ and $\boldsymbol{w}^*$ using the approximation error of the kernel matrix, where $f^*$ is the optimal hypothesis in the original RKHS in hindsight, and $\boldsymbol{w}^*$ is the optimal hypothesis on the incremental randomized sketches in hindsight. Specifically, we utilize the following conclusion from Theorem 2 in (Yang et al., 2012):

$$\ell(\boldsymbol{w}^*) - \ell(f^*) \leq \frac{1}{2T\lambda} \|\boldsymbol{K}_{\text{sk}}^{(T)} - \boldsymbol{K}\|_2,$$

yielding that

$$\sum_{t=1}^{T} (\ell_t(\boldsymbol{w}^*) - \ell_t(f^*)) \leq \frac{1}{2\lambda} \left( \left\| [\boldsymbol{K}_{\text{sk}}^{(T)}]_{B,\rho} - \boldsymbol{K}_{B,\rho} \right\|_2 + \left\| \widehat{\boldsymbol{K}}_{B,\rho} - \boldsymbol{K} \right\|_2 \right) \tag{13}$$

where $\boldsymbol{K}_{B,\rho} \in \mathbb{R}^{(B+\lfloor (T-B)/\rho \rfloor) \times (B+\lfloor (T-B)/\rho \rfloor)}$ is the intersection matrix of $\boldsymbol{K}$, constructed using $B + \lfloor (T-B)/\rho \rfloor$ examples, $[\boldsymbol{K}\text{sk}^{(T)}]B,\rho$ is the approximate matrix for $\boldsymbol{K}_{B,\rho}$ obtained using the proposed incremental sketching method with a rank parameter $k$, $\boldsymbol{O}$ is a zero matrix of size $(T - B - \lfloor (T-B)/\rho \rfloor) \times (T - B - \lfloor (T-B)/\rho \rfloor)$, and

$$\widehat{\boldsymbol{K}}_{B,\rho} = \text{diag}\, \{\boldsymbol{K}_{B,\rho},\, \boldsymbol{O}\} \in \mathbb{R}^{T \times T},$$

$$\widehat{[\boldsymbol{K}_{\text{sk}}^{(T)}]}_{B,\rho} = \text{diag}\, \left\{ [\boldsymbol{K}_{\text{sk}}^{(T)}]_{B,\rho},\, \boldsymbol{O} \right\} \in \mathbb{R}^{T \times T}.$$

Given that the eigenvalues of the kernel matrix decay polynomially with a decay rate $\beta > 1$, we can establish the following bound:

$$
\begin{aligned}
\|\widehat{\boldsymbol{K}}_{B,\rho} - \boldsymbol{K}\|_2 &\leq \frac{T - B - \lfloor (T-B)/\rho \rfloor}{T} \sum_{i=1}^{T} i^{-\beta} \\
&\leq \frac{T - B - \lfloor (T-B)/\rho \rfloor}{T} \int_1^T i^{-\beta} \mathrm{d}i \\
&= \frac{T - B - \lfloor (T-B)/\rho \rfloor}{T} \frac{1}{\beta - 1} \left( 1 - \frac{1}{T^{\beta-1}} \right) \\
&\leq \frac{1}{\beta - 1} \left( 1 - \frac{B + \lfloor (T-B)/\rho \rfloor}{T} \right).
\end{aligned}
\tag{14}
$$

Besides, from Assumption 3, with probability at least $1 - \delta$, we have

$$
\left\| [\boldsymbol{K}_{\mathrm{sk}}^{(T)}]_{B,\rho} - \boldsymbol{K}_{B,\rho} \right\|_2 \leq \sqrt{1+\epsilon} \, \left\| [\boldsymbol{C}_{\mathrm{m}} \boldsymbol{F}_{\mathrm{mod}} \boldsymbol{C}_{\mathrm{m}}^{\mathsf{T}}]_{B,\rho} - \boldsymbol{K}_{B,\rho} \right\|_{\mathrm{F}},
\tag{15}
$$

where $[\boldsymbol{C}_{\mathrm{m}} \boldsymbol{F}_{\mathrm{mod}} \boldsymbol{C}_{\mathrm{m}}^{\mathsf{T}}]_{B,\rho}$ is the approximate matrix for $\boldsymbol{K}_{B,\rho}$ using the modified Nyström approach with a rank parameter $k$.

Denoting the best rank-$k$ approximation of $\boldsymbol{A}$ as $(\boldsymbol{A})_k$, and considering that the eigenvalues of $\boldsymbol{K}$ decay polynomially with a decay rate $\beta > 1$, we can find a value of $\beta > 1$ such that $\lambda_i(\boldsymbol{K}) = O(i^{-\beta})$. This leads to the following expression:

$$
\|\boldsymbol{K}_{B,\rho} - (\boldsymbol{K}_{B,\rho})_k\|_{\mathrm{F}} = \sqrt{B + \lfloor (T-B)/\rho \rfloor - k} \cdot (k+1)^{-\beta} = O(\sqrt{B}).
\tag{16}
$$

Given $\epsilon' \in (0,1)$, when $s_{\mathrm{m}} = \Omega(\mu(\boldsymbol{K}_{B,\rho})k \log k)$, according to Theorem 22 in (Wang et al., 2016), we can derive the following bound:

$$
\begin{aligned}
&\|[\boldsymbol{C}_{\mathrm{m}} \boldsymbol{F}_{\mathrm{mod}} \boldsymbol{C}_{\mathrm{m}}^{\mathsf{T}}]_{B,\rho} - \boldsymbol{K}_{B,\rho}\|_{\mathrm{F}} \\
&\leq \left\| [\boldsymbol{C}_{\mathrm{m}} \boldsymbol{F}_{\mathrm{mod}} \boldsymbol{C}_{\mathrm{m}}^{\mathsf{T}}]_{B,\rho} - [\boldsymbol{C}_{\mathrm{m}} \boldsymbol{C}_{\mathrm{m}}^{\dagger} \boldsymbol{K}_{B,\rho}]_{B,\rho} \right\|_{\mathrm{F}} + \left\| [\boldsymbol{C}_{\mathrm{m}} \boldsymbol{C}_{\mathrm{m}}^{\dagger} \boldsymbol{K}_{B,\rho}]_{B,\rho} - \boldsymbol{K}_{B,\rho} \right\|_{\mathrm{F}} \\
&= \left\| [\boldsymbol{C}_{\mathrm{m}} \boldsymbol{C}_{\mathrm{m}}^{\dagger} \boldsymbol{K}_{B,\rho} \left( \boldsymbol{C}_{\mathrm{m}} \boldsymbol{C}_{\mathrm{m}}^{\dagger} \right)^{\mathsf{T}} - \boldsymbol{C}_{\mathrm{m}} \boldsymbol{C}_{\mathrm{m}}^{\dagger} \boldsymbol{K}_{B,\rho}]_{B,\rho} \right\|_{\mathrm{F}} + \left\| [\boldsymbol{C}_{\mathrm{m}} \boldsymbol{C}_{\mathrm{m}}^{\dagger} \boldsymbol{K}_{B,\rho}]_{B,\rho} - \boldsymbol{K}_{B,\rho} \right\|_{\mathrm{F}} \\
&\leq \left\| [\boldsymbol{C}_{\mathrm{m}} \boldsymbol{C}_{\mathrm{m}}^{\dagger}]_{B,\rho} \right\|_{\mathrm{F}} \left\| [\boldsymbol{K}_{B,\rho} \left( \boldsymbol{C}_{\mathrm{m}} \boldsymbol{C}_{\mathrm{m}}^{\dagger} \right)^{\mathsf{T}}]_{B,\rho} - \boldsymbol{K}_{B,\rho} \right\|_{\mathrm{F}} + \left\| [\boldsymbol{C}_{\mathrm{m}} \boldsymbol{C}_{\mathrm{m}}^{\dagger} \boldsymbol{K}_{B,\rho}]_{B,\rho} - \boldsymbol{K}_{B,\rho} \right\|_{\mathrm{F}} \\
&= \left( 1 + \left\| [\boldsymbol{C}_{\mathrm{m}} \boldsymbol{C}_{\mathrm{m}}^{\dagger}]_{B,\rho} \right\|_{\mathrm{F}} \right) \left\| [\boldsymbol{C}_{\mathrm{m}} \boldsymbol{C}_{\mathrm{m}}^{\dagger} \boldsymbol{K}_{B,\rho}]_{B,\rho} - \boldsymbol{K}_{B,\rho} \right\|_{\mathrm{F}} \\
&\leq \left( 1 + \sqrt{s_{\mathrm{m}}} \right) \left\| [\boldsymbol{C}_{\mathrm{m}} \boldsymbol{C}_{\mathrm{m}}^{\dagger} \boldsymbol{K}_{B,\rho}]_{B,\rho} - \boldsymbol{K}_{B,\rho} \right\|_{\mathrm{F}} \\
&\leq \sqrt{1 + \epsilon'} \left( 1 + \sqrt{s_{\mathrm{m}}} \right) \|\boldsymbol{K}_{B,\rho} - (\boldsymbol{K}_{B,\rho})_k\|_F,
\end{aligned}
\tag{17}
$$

where $[\boldsymbol{A}]B,\rho$ indicates that $\boldsymbol{A}$ is constructed based on the matrix $\boldsymbol{K}B,\rho$, and $\mu(\boldsymbol{K}_{B,\rho})$ represents the coherence of $\boldsymbol{K}_{B,\rho}$. By combining equation 15, equation 16, and equation 17, we obtain the following result:

$$
\left\| [\boldsymbol{K}_{\mathrm{sk}}^{(T)}]_{B,\rho} - \boldsymbol{K}_{B,\rho} \right\|_2 \leq \sqrt{1+\epsilon} \, O(\sqrt{B}).
\tag{18}
$$

Substituting equation 14 and equation 18 into equation 13, we have

$$
\begin{aligned}
&\sum_{t=1}^{T} (\ell_t(\boldsymbol{w}^*) - \ell_t(f^*)) \\
&\leq \frac{1}{2\lambda(\beta - 1)} \left( 1 - \frac{B + \lfloor (T-B)/\rho \rfloor}{T} \right) + \frac{\sqrt{1+\epsilon}}{2\lambda} O(\sqrt{B}) + \frac{\lambda}{2} \|f^*\|_{\mathcal{H}_\kappa}^2 - \frac{\lambda}{2} \|\boldsymbol{w}^*\|_2^2.
\end{aligned}
\tag{19}
$$

Next, we analyze the regret resulting from hypothesis updating on the incremental randomized sketches. We begin by decomposing $\ell_t(\boldsymbol{w}_t) - \ell_t(\boldsymbol{w}^*)$ into two terms as follows:

$$
\ell_t(\boldsymbol{w}_t) - \ell_t(\boldsymbol{w}^*) = \underbrace{\ell_t(\boldsymbol{w}_t) - \ell_t(\boldsymbol{w}_t^*)}_{\text{Term 1: Optimization Error}} + \underbrace{\ell_t(\boldsymbol{w}_t^*) - \ell_t(\boldsymbol{w}^*)}_{\text{Term 2: Estimation Error}},
$$

where $f_t^*(\cdot) = \langle \boldsymbol{w}_t^*, \boldsymbol{\phi}_t(\cdot) \rangle$ represents the optimal hypothesis on the incremental sketches for the first $t$ instances, and $\boldsymbol{w}^*$ denotes the optimal hypothesis on the incremental sketches in hindsight.

The optimization error quantifies the discrepancy between the hypothesis generated by the proposed faster second-order online kernel learning algorithm and the optimal hypothesis on the incremental randomized sketches at each round. On the other hand, the estimation error measures the difference between the optimal hypotheses on the incremental randomized sketches for the first $t$ instances and for all $T$ instances, respectively.

To obtain an upper bound for the optimization error, we leverage the directional curvature condition presented in Assumption 2. Given that the Euclidean regularization is a strongly convex regularizer, the loss function $\ell_t$ also satisfies the directional curvature condition. As a result, we can utilize the inequality provided in Assumption 2 to bound the optimization error. Specifically, we obtain the following expression:

$$\ell_t(\boldsymbol{w}_t) - \ell_t(\boldsymbol{w}_t^*) \leq \langle \nabla \ell_t(\boldsymbol{w}_t), \boldsymbol{w}_t - \boldsymbol{w}_t^* \rangle - \frac{L_{\text{Cur}}}{2} \langle \nabla \ell_t(\boldsymbol{w}_t), \boldsymbol{w}_t^* - \boldsymbol{w}_t \rangle^2 . \tag{20}$$

Letting

$$\Delta_t = \langle \nabla \ell_t(\boldsymbol{w}_t), \boldsymbol{w}_t - \boldsymbol{w}_t^* \rangle - \frac{L_{\text{Cur}}}{2} \langle \nabla \ell_t(\boldsymbol{w}_t), \boldsymbol{w}_t^* - \boldsymbol{w}_t \rangle^2 ,$$

equation 20 can be rewritten as $\ell_t(\boldsymbol{w}_t) - \ell_t(\boldsymbol{w}_t^*) \leq \Delta_t$. Note that $\boldsymbol{g}_t = \nabla \ell_t(\boldsymbol{w}_t)$ in the FORKS algorithm, we first give the bound of $\langle \boldsymbol{g}_t, \boldsymbol{w}_t - \boldsymbol{w}_t^* \rangle = \langle \nabla \ell_t(\boldsymbol{w}_t), \boldsymbol{w}_t - \boldsymbol{w}_t^* \rangle$ in $\Delta_t$. Based on the update steps for $\boldsymbol{v}_t$ and $\boldsymbol{w}_t$ proposed in FORKS, it can be inferred that

$$\boldsymbol{v}_{t+1} - \boldsymbol{w}_t^* = \boldsymbol{w}_t - \boldsymbol{w}_t^* - \boldsymbol{A}_t^{-1} \boldsymbol{g}_t, \quad \boldsymbol{A}_t(\boldsymbol{v}_{t+1} - \boldsymbol{w}_t^*) = \boldsymbol{A}_t(\boldsymbol{w}_t - \boldsymbol{w}_t^*) - \boldsymbol{g}_t,$$

yielding that

$$\begin{aligned} &\langle \boldsymbol{v}_{t+1} - \boldsymbol{w}_t^*, \boldsymbol{A}_t(\boldsymbol{v}_{t+1} - \boldsymbol{w}_t^*) \rangle \\ &= \langle \boldsymbol{w}_t - \boldsymbol{w}_t^*, \boldsymbol{A}_t(\boldsymbol{w}_t - \boldsymbol{w}_t^*) \rangle - 2 \langle \boldsymbol{g}_t, \boldsymbol{w}_t - \boldsymbol{w}_t^* \rangle + \langle \boldsymbol{g}_t, \boldsymbol{A}_t^{-1} \boldsymbol{g}_t \rangle. \end{aligned} \tag{21}$$

Considering that $\boldsymbol{w}_{t+1}$ in FORKS can be interpreted as the generalized projection of $\boldsymbol{v}_{t+1}$ within the norm induced by $\boldsymbol{A}_t$, by leveraging equation 21 and the Pythagorean theorem, we can derive the following relationship:

$$\begin{aligned} &2 \langle \boldsymbol{g}_t, \boldsymbol{w}_t - \boldsymbol{w}_t^* \rangle \\ &= \langle \boldsymbol{w}_t - \boldsymbol{w}_t^*, \boldsymbol{A}_t(\boldsymbol{w}_t - \boldsymbol{w}_t^*) \rangle + \langle \boldsymbol{g}_t, \boldsymbol{A}_t^{-1} \boldsymbol{g}_t \rangle - \langle \boldsymbol{v}_{t+1} - \boldsymbol{w}_t^*, \boldsymbol{A}_t(\boldsymbol{v}_{t+1} - \boldsymbol{w}_t^*) \rangle \\ &\leq \langle \boldsymbol{w}_t - \boldsymbol{w}_t^*, \boldsymbol{A}_t(\boldsymbol{w}_t - \boldsymbol{w}_t^*) \rangle + \langle \boldsymbol{g}_t, \boldsymbol{A}_t^{-1} \boldsymbol{g}_t \rangle - \langle \boldsymbol{w}_{t+1} - \boldsymbol{w}_t^*, \boldsymbol{A}_t(\boldsymbol{w}_{t+1} - \boldsymbol{w}_t^*) \rangle. \end{aligned} \tag{22}$$

By summing equation 22 for $t \in [T]$, combining with equation 20 we obtain

$$\begin{aligned} &\sum_{t=1}^{T} \ell_t(\boldsymbol{w}_t) - \ell_t(\boldsymbol{w}_t^*) \\ &\leq \sum_{t=1}^{T} \langle \boldsymbol{g}_t, \boldsymbol{w}_t - \boldsymbol{w}_t^* \rangle - \sum_{t=1}^{T} \frac{L_{\text{Cur}}}{2} \langle \nabla \ell_t(\boldsymbol{w}_t), \boldsymbol{w}_t^* - \boldsymbol{w}_t \rangle^2 \\ &\leq \frac{1}{2} \sum_{t=1}^{T} \langle \boldsymbol{w}_t - \boldsymbol{w}_t^*, \boldsymbol{A}_t(\boldsymbol{w}_t - \boldsymbol{w}_t^*) \rangle + \frac{1}{2} \sum_{t=1}^{T} \langle \boldsymbol{g}_t, \boldsymbol{A}_t^{-1} \boldsymbol{g}_t \rangle - \\ &\quad \frac{1}{2} \sum_{t=1}^{T} \langle \boldsymbol{w}_{t+1} - \boldsymbol{w}_t^*, \boldsymbol{A}_t(\boldsymbol{w}_{t+1} - \boldsymbol{w}_t^*) \rangle - \sum_{t=1}^{T} \frac{L_{\text{Cur}}}{2} \langle \nabla \ell_t(\boldsymbol{w}_t), \boldsymbol{w}_t^* - \boldsymbol{w}_t \rangle^2 . \end{aligned} \tag{23}$$

Since incremental sketches in FORKS are periodically updated, $\boldsymbol{w}_t^*$ can be updated at most $\lfloor (T - B)/\rho \rfloor$ times. Consequently, by leveraging the fact that $\boldsymbol{A}_{t+1} = \boldsymbol{A}_t + \sigma_t \boldsymbol{g}_t \boldsymbol{g}_t^\top$, where $\sigma_t \geq L_{\text{Cur}}$,

the upper bound in equation 23 can be simplified to the following expression:

$$\sum_{t=1}^{T} \ell_t(\boldsymbol{w}_t) - \ell_t(\boldsymbol{w}_t^*)$$

$$\leq \frac{1}{2}\langle \boldsymbol{w}_1 - \boldsymbol{w}_1^*, (\boldsymbol{A}_2 - \boldsymbol{g}_1\boldsymbol{g}_1^\top/2)(\boldsymbol{w}_1 - \boldsymbol{w}_1^*)\rangle + \frac{1}{2}\sum_{t=1}^{T}\langle \boldsymbol{g}_t, \boldsymbol{A}_t^{-1}\boldsymbol{g}_t\rangle +$$

$$\frac{1}{2}\sum_{t=1}^{T}(\boldsymbol{w}_t - \boldsymbol{w}_t^*)^\top (\boldsymbol{A}_t - \boldsymbol{A}_{t-1} - \sigma_t\boldsymbol{g}_t\boldsymbol{g}_t^\top)(\boldsymbol{w}_t - \boldsymbol{w}_t^*)$$

$$= \frac{1}{2}\langle \boldsymbol{w}_1 - \boldsymbol{w}_1^*, \boldsymbol{A}_1(\boldsymbol{w}_1 - \boldsymbol{w}_1^*)\rangle + \frac{1}{2}\sum_{t=1}^{T}\langle \boldsymbol{g}_t, \boldsymbol{A}_t^{-1}\boldsymbol{g}_t\rangle + \qquad (24)$$

$$\sum_{t=1}^{T}\frac{\eta_t}{2}(\boldsymbol{w}_t - \boldsymbol{w}_t^*)^\top \boldsymbol{g}_t\boldsymbol{g}_t^\top (\boldsymbol{w}_t - \boldsymbol{w}_t^*)$$

$$= \frac{\alpha}{2}\|\boldsymbol{w}_1 - \boldsymbol{w}_1^*\|_2^2 + \frac{1}{2}\sum_{t=1}^{T}\langle \boldsymbol{g}_t, \boldsymbol{A}_t^{-1}\boldsymbol{g}_t\rangle$$

$$\leq \frac{\alpha D_{\boldsymbol{w}}^2}{2} + \frac{1}{2}\sum_{t=1}^{T}\langle \boldsymbol{g}_t, \boldsymbol{A}_t^{-1}\boldsymbol{g}_t\rangle,$$

By applying the result from (Hazan et al., 2007), we can obtain the following upper bound on the sum of the inner products $\langle \boldsymbol{g}_t, \boldsymbol{A}_t^{-1}\boldsymbol{g}_t\rangle, \forall t \in [T]$:

$$\sum_{t=1}^{T}\langle \boldsymbol{g}_t, \boldsymbol{A}_t^{-1}\boldsymbol{g}_t\rangle \leq \frac{1}{L_{\mathrm{Cur}}}\log\left(TL_{\mathrm{Lip}}^2/L_{\mathrm{Cur}} + 1\right)^k = \frac{k}{L_{\mathrm{Cur}}}\log\left(TL_{\mathrm{Lip}}^2/L_{\mathrm{Cur}} + 1\right). \qquad (25)$$

Combining equation 25 with equation 24, we find that

$$\sum_{t=1}^{T}(\ell_t(\boldsymbol{w}_t) - \ell_t(\boldsymbol{w}_t^*))$$

$$\leq \frac{\alpha D_{\boldsymbol{w}}^2}{2} + \frac{k}{2L_{\mathrm{Cur}}}O(\log T) + \frac{\lambda}{2}\|\boldsymbol{w}_t^*\|_2^2 - \frac{\lambda}{2}\|\boldsymbol{w}_t\|_2^2. \qquad (26)$$

For the estimation error, we obtain the following upper bound

$$\sum_{t=1}^{T}(\ell_t(\boldsymbol{w}_t^*) - \ell_t(\boldsymbol{w}^*))$$

$$\leq \frac{1}{2\lambda}\left\|\boldsymbol{K}_{\mathrm{sk}}^{(T_0)} - \boldsymbol{K}_{\mathrm{sk}}^{(T)}\right\|_2 + \frac{\lambda}{2}\|\boldsymbol{w}^*\|_2^2 - \frac{\lambda}{2}\|\boldsymbol{w}_t^*\|_2^2 \qquad (27)$$

$$\leq \frac{1}{2\lambda}\left(\left\|\boldsymbol{K}_{\mathrm{sk}}^{(T_0)} - \boldsymbol{K}^{(T_0)}\right\|_2 + \left\|\boldsymbol{K}^{(T_0)} - \boldsymbol{K}\right\|_2 + \left\|\boldsymbol{K}_{\mathrm{sk}}^{(T)} - \boldsymbol{K}\right\|_2\right) + \frac{\lambda}{2}\|\boldsymbol{w}^*\|_2^2 - \frac{\lambda}{2}\|\boldsymbol{w}_t^*\|_2^2$$

$$\leq \frac{1}{2\lambda}\left[\sqrt{1+\tilde{\epsilon}}\,O(\sqrt{B}) + \frac{1}{\beta - 1}\left(1 - \frac{B}{T}\right) + \left\|\boldsymbol{K}_{\mathrm{sk}}^{(T)} - \boldsymbol{K}\right\|_2\right] + \frac{\lambda}{2}\|\boldsymbol{w}^*\|_2^2 - \frac{\lambda}{2}\|\boldsymbol{w}_t^*\|_2^2.$$

Finally, the three inequalities equation 19, equation 26 and equation 27 combined give the following bound:

$$\sum_{t=1}^{T}(\ell_t(\boldsymbol{w}_t) - \ell_t(f^*))$$

$$\leq \frac{\alpha D_{\boldsymbol{w}}^2}{2} + \frac{k}{2L_{\mathrm{Cur}}}O(\log T) +$$

$$\frac{\lambda}{2}\|f^*\|_{\mathcal{H}_\kappa}^2 + \frac{1}{\lambda(\beta - 1)}\left(\frac{3}{2} - \frac{B + \lfloor(T-B)/\rho\rfloor}{T}\right) + \frac{\sqrt{1+\epsilon}}{\lambda}O(\sqrt{B}).$$

$$\square$$

# G  PROOF OF REMARK 2

*Proof.* By leveraging the fact that $\boldsymbol{A}_{t+1} = \boldsymbol{A}_t + (\sigma_t + \eta_t)\boldsymbol{g}_t\boldsymbol{g}_t^\top$, where $\sigma_t \geq L_{\mathrm{Cur}}$ and applying the Proposition 1 from (Luo et al., 2016), we can rewrite equation 24 as:

$$
\begin{aligned}
&\sum_{t=1}^{T} \ell_t(\boldsymbol{w}_t) - \ell_t(\boldsymbol{w}_t^*) \\
&\leq \frac{1}{2}\langle \boldsymbol{w}_1 - \boldsymbol{w}_1^*, (\boldsymbol{A}_2 - \boldsymbol{g}_1\boldsymbol{g}_1^\top/2)(\boldsymbol{w}_1 - \boldsymbol{w}_1^*)\rangle + \frac{1}{2}\sum_{t=1}^{T}\langle \boldsymbol{g}_t, \boldsymbol{A}_t^{-1}\boldsymbol{g}_t\rangle + \\
&\quad \frac{1}{2}\sum_{t=1}^{T}(\boldsymbol{w}_t - \boldsymbol{w}_t^*)^\top(\boldsymbol{A}_t - \boldsymbol{A}_{t-1} - \sigma_t\boldsymbol{g}_t\boldsymbol{g}_t^\top)(\boldsymbol{w}_t - \boldsymbol{w}_t^*) \\
&= \frac{1}{2}\langle \boldsymbol{w}_1 - \boldsymbol{w}_1^*, \boldsymbol{A}_1(\boldsymbol{w}_1 - \boldsymbol{w}_1^*)\rangle + \frac{1}{2}\sum_{t=1}^{T}\langle \boldsymbol{g}_t, \boldsymbol{A}_t^{-1}\boldsymbol{g}_t\rangle + \\
&\quad \sum_{t=1}^{T}\frac{\eta_t}{2}(\boldsymbol{w}_t - \boldsymbol{w}_t^*)^\top\boldsymbol{g}_t\boldsymbol{g}_t^\top(\boldsymbol{w}_t - \boldsymbol{w}_t^*) \\
&= \frac{\alpha}{2}\|\boldsymbol{w}_1 - \boldsymbol{w}_1^*\|_2^2 + \frac{1}{2}\sum_{t=1}^{T}\langle \boldsymbol{g}_t, \boldsymbol{A}_t^{-1}\boldsymbol{g}_t\rangle + \sum_{t=1}^{T}\frac{\eta_t}{2}(\boldsymbol{w}_t - \boldsymbol{w}_t^*)^\top\boldsymbol{g}_t\boldsymbol{g}_t^\top(\boldsymbol{w}_t - \boldsymbol{w}_t^*) \\
&\leq \frac{\alpha D_{\boldsymbol{w}}^2}{2} + \frac{1}{2}\sum_{t=1}^{T}\langle \boldsymbol{g}_t, \boldsymbol{A}_t^{-1}\boldsymbol{g}_t\rangle + 2L_{\mathrm{Lip}}^2\sum_{t=1}^{T}\eta_t \\
&\leq \frac{\alpha D_{\boldsymbol{w}}^2}{2} + \frac{k}{2(\eta_T + L_{\mathrm{Cur}})}O(\log T) + 2L_{\mathrm{Lip}}^2\sum_{t=1}^{T}\eta_t,
\end{aligned}
\tag{28}
$$

In the worst case, if $L_{\mathrm{Cur}} = 0$, we set $\eta_t = \sqrt{\frac{k}{L_{\mathrm{Lip}}^2 t}}$ and the bound can be simplified to:

$$
\begin{aligned}
&\sum_{t=1}^{T}(\ell_t(\boldsymbol{w}_t) - \ell_t(f^*)) \\
&\leq \frac{\alpha D_{\boldsymbol{w}}^2}{2} + \frac{\sqrt{k}L_{\mathrm{Lip}}}{2}O(\sqrt{T}) + 4\sqrt{k}L_{\mathrm{Lip}}O(\sqrt{T}) \\
&\quad \frac{\lambda}{2}\|f^*\|_{\mathcal{H}_\kappa}^2 + \frac{1}{\lambda(\beta - 1)}\left(\frac{3}{2} - \frac{B + \lfloor(T - B)/\rho\rfloor}{T}\right) + \frac{\sqrt{1 + \epsilon}}{\lambda}O(\sqrt{B}).
\end{aligned}
$$

□

# H  DATASET AND EXPERIMENTAL SETUP

We evaluate FORKS on several real-world datasets for binary classification tasks. We use several well-known classification benchmarks for online learning, where the number of instances ranges from 1000 to $581{,}012$. All the experiments are performed over 20 different random permutations of the datasets. Besides, we introduce a large-scale real-world dataset `KuaiRec` (Gao et al., 2022), which has $4{,}494{,}578$ instances and associated timestamps. We do not tune the stepsizes $\eta$ of all the gradient descent-based algorithms but take the value $\eta = 0.2$. We uniformly set $d = 1$, $\alpha = 0.01$, $\eta_i = 0$, $\sigma_i = 0.5$ and $\lambda = 0.01$ for FORKS and SkeGD. We take the Gaussian kernel $\kappa(\boldsymbol{x}_i, \boldsymbol{x}_j) = \exp\left(\frac{-\|\boldsymbol{x}_i - \boldsymbol{x}_j\|_2^2}{2\sigma^2}\right)$ with parameter set $\sigma \in \{2^{[-5:+0.5:7]}\}$ used by Zhang & Liao (2019). All experiments are performed on a machine with 24-core Intel(R) Xeon(R) Gold 6240R 2.40GHz CPU and 256 GB memory.

## I   MORE ABOUT KUAIREC DATASET

For our experiment, we utilize KuaiRec's small matrix as the dataset. The processing of dependent variables involves dividing the ratio of the user's time spent on the video to the video duration (watch ratio) by a threshold of 0.75, with values greater than 0.75 classified as positive and values less than or equal to 0.75 classified as negative. The selection of independent variables is obtained from three csv files, as specified in the code.

## J   ADDITIONAL EXPERIMENT RESULTS

### J.1   ADDITIONAL EXPERIMENT RESULTS UNDER ADVERSARIAL ENVIRONMENT

We set $b = 500$, $r = 20$ for `codrna-2` and `german-2`. The results are presented in Table 4.

Table 4: Comparisons among BOGD, NOGD, PROS-N-KONS, SkeGD and our FORKS w.r.t. the mistake rates (%) and the running time (s). The best result is highlighted in **bold** font.

| Algorithm | codrna-2 | | german-2 | |
|---|---|---|---|---|
| | Mistake rate | Time | Mistake rate | Time |
| BOGD | $14.745 \pm 0.063$ | 0.043 | $21.290 \pm 0.918$ | 0.060 |
| NOGD | $19.977 \pm 1.536$ | 0.041 | $16.527 \pm 0.810$ | 0.056 |
| PROS-N-KONS | $15.430 \pm 2.315$ | 20.612 | $11.187 \pm 1.782$ | 1.144 |
| SkeGD | $15.829 \pm 2.583$ | 0.203 | $5.742 \pm 2.647$ | 0.077 |
| FORKS | $\mathbf{4.127 \pm 0.769}$ | 0.039 | $\mathbf{2.960 \pm 0.185}$ | 0.050 |

We demonstrate that all methods exhibit improved performance in a less hostile adversarial setting. Nevertheless, FORKS remains superior to other algorithms with significant advantages in terms of both time and prediction performance.

### J.2   ADDITIONAL EXPERIMENT RESULTS UNDER LARGE-SCALE REAL-WORLD DATASETS

Similar to the experimental setup described in section 5.3, we evaluate the performance of the algorithms across various budgets $B$, spanning from 100 to 500. To avoid excessive training time, we use a budgeted version of PROS-N-KONS with a maximum budget of $B_{\max} = 100$. Since the buffer size of PROS-N-KONS is data-dependent, we repeat the training process 20 times to compute the average error rate and the logarithm of the average time for comparison.

From Figure 4, we can observe that our FORKS show the best learning performance under most budget conditions. The large-scale experiments validate the effectiveness and efficiency of our proposed FORKS, making it potentially more practical than the existing second-order online kernel learning approaches. Meanwhile, we observed that increasing the budget size $B$ results in a lower mistake rate but also leads to a higher computation time cost. In practice, we can flexibly adjust the budget size based on our estimates of the data stream size to obtain a better approximation quality.

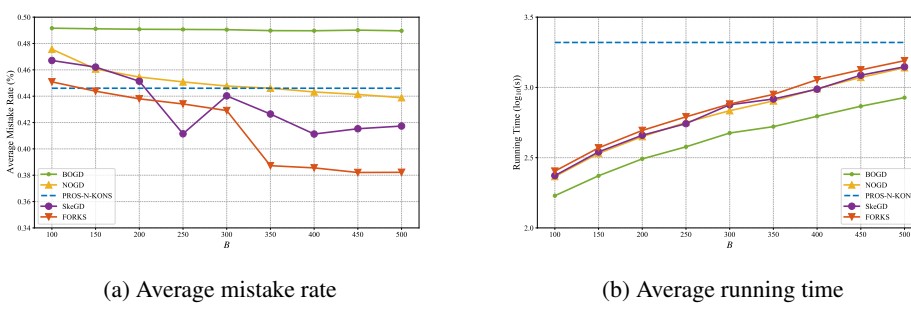

(a) Average mistake rate                    (b) Average running time

Figure 4: The mistake rates and average running time on `KuaiRec` under hinge loss. As PROS-N-KONS utilizes an adaptive budget, it cannot modify computational costs, thereby being depicted as a parallel line in the figures.

