# OpenReview forum: "FORKS: Fast Second-Order Online Kernel Learning using Incremental Sketching"
_ICLR.cc/2024/Conference — Submitted to ICLR 2024_

### Official Review · Reviewer_jbwL · 2023-10-13

**Soundness:** 3 good
**Presentation:** 2 fair
**Contribution:** 2 fair
**Rating:** 5
**Confidence:** 2

**Summary:**

This paper considers how to speed up the second-order method for online kernel learning using sketching. To achieve a logarithmic regret bound, it is common to deploy Newton-type method for OKL, but second-order methods are usually computationally inefficient. This paper attempts to address the issue by using sketching and better algorithms. To this end, they provide an algorithm with update time $O(Bk+k^2)$, when $k$ is small this is a strict improvement from prior works that rely on computing the Newton direction exactly. They also conduct extensive experiments to show that the proposed method is faster than other second-order methods and has a better error rate than first-order methods.

**Strengths:**

The algorithm in this paper is relatively simple and elegant. It uses sketching in a natural and unsurprising way to improve the runtime efficiency. It also proposes a decomposition algorithm that can effectively handle low-rank updates, by computing a truncated SVD on a much smaller matrix. This leads to the overall runtime improvements. These improvements are also visible through experiments.

**Weaknesses:**

The idea of using sketching to speed up second-order methods are not novel, for example, the Newton sketch due to Pilanci and Wainwright and the spectral approximation approach in several papers on training neural networks [BPSW21]. Other than the sketching speedup, and the truncated SVD procedure, this paper follows a very standard pipeline to analyze the performance of their modified algorithm. Overall, the results are not surprising and the novelty is limited.

[BPSW21] J. van den Brand, B. Peng, Z. Song and O. Weinstein. Training (over-parametrized) neural networks in nearly linear time. ITCS'21.

**Questions:**

Typo: on page 3, two paragraphs above section 3.2, $K_{i,j}$ should be $\phi(x_i)^\top \phi(x_j)$ instead of $\phi(x_i)\phi(x_j)^\top$.

---

### Official Review · Reviewer_9VaC · 2023-11-01

**Soundness:** 3 good
**Presentation:** 3 good
**Contribution:** 3 good
**Rating:** 6
**Confidence:** 4

**Summary:**

This paper propose FORKS:Fast Second-Order Online Kernel Learning using Incremental Sketching, a fast and effective second-order online kernel learning method. FORKS use efficient low-rank modifications to maintain incremental randomized sketches and apply it to second-order online kernel learning, which makes FORKS have linear time complexity w.r.t. the budget, and enjoy a logarithmic regret bound for online kernel learning.
The proposed TISVD : Truncated Incremental SVD, is a novel incremental singular value decomposition based method adapting to matrix decomposition problems.
The paper conducts an extensive experimental study to demonstrate the superior performance of FORKS on several datasets and validate the robustness and scalability of FORKS on large-scale datasets.

**Strengths:**

Originality:It proposed a new incremental SVD method,TISVD. It simplifies the singular value decomposition of the updated matrix to a singular value decomposition of a matrix with lower rank,which is very effective in maintaining the incremental maintenance of kernel matrices
Quality: This paper combines the strengths of two current articles in the field of online kernel learning and proposes a new framework FORKS, which greatly improves the time efficiency and accuracy.
Clarity；This paper systematically elaborates on the steps of its framework , and analyzes the complexity and the regret bound of FORKS.The proposed framework is easy to follow.
Significance:This proposed framework significantly improves the efficiency of each incremental update step while ensuring the logarithmic regret bound.

**Weaknesses:**

The original content of this paper is relatively limited. The main framework of this paper is a simple combination of important parts from the other two papers ,SkeGD and PRONS-N-KONS.
In addition, stacking formulas simply is unfriendly to those who are not familiar with incremental random sketch,and more images would be better.

**Questions:**

1.Compared with other ISVD methods, TISVD performs a truncated rank SVD. Will there be any loss of information in the original matrix, as low rank is one of the prerequisites for ISVD?

---

> ### Author Response · Authors · 2023-11-18
>
> **Response to Reviewer 9VaC**:
>
> Thanks for your detailed review of our work!
>
> **Reply to Weaknesses**:
>
> The truncated rank step will induce a loss of information in the original matrix. However, according to the experiment, the loss of information is limited.

---

### Official Review · Reviewer_SDsQ · 2023-11-02

**Soundness:** 1 poor
**Presentation:** 2 fair
**Contribution:** 2 fair
**Rating:** 1
**Confidence:** 4

**Summary:**

This paper considers kernel learning in the online setting. The main goal is to give an algorithm with logarithmic regret that is also efficient. It is well known that online gradient descent based methods only achieve square root regret, while exact second order optimization methods achieve logarithmic regret. Thus,  the paper proposes a new approximate second order method, which approximates the kernel matrix at any time step using an online variant of CUR Decomposition/Nystrom approximation, proposed by Wang et al. 2016. This method is similar to others that have been proposed in the literature that use Nystrom based kernel matrix approximation to speed up second second order online kernel learning.

Naively, applying this Nystrom decomposition method requires time roughly s^3 per iteration where s is the number of landmark points sampled. This is due to having to perform a truncated SVD of an s x s sampled matrix to reduce it to some rank k < s.  To address this, the paper proposes an incremental SVD approach which improves the s^3 runtime to O(sk+k^3).

The paper provides a regret analysis for their approach, as well as experiments. The paper also claims that the proposed algorithm is more robust in adversarial settings than existing approaches. However, it was unclear to me why this was — given that the algorithm largely follows the same blueprint as related prior work. No guarantees on robustness are given.

I have two related major concerns about the presented results.

1. The relationship between the proposed truncated incremental SVD (TISVD), (which seems to be the main algorithmic contribution of the work) and existing work on incremental SVD was unclear to me. Existing work on incremental SVD updates the SVD of a s x s rank-k matrix with a constant rank update in O(sk+k^3) time. Once this update is performed, the matrix may no-longer be rank-k. However, one can trivially truncate the new SVD back to rank-k if desired. This seems to be the only difference proposed by TISVD? It would be helpful if the authors could explain why the given O(sk+k^3) runtime then just doesn’t directly follow from existing work on Incremental SVD. There is an extensive discussion in appendix D, but it is not clear and does not address my question above.

2. Of course, truncating the SVD after each low-rank update, as is done in this paper has a cost. It means that after all the updates have been performed the output *is not* equivalent to just performing a truncated SVD on the sum of updates. Consider a simple example where I have a diagonal 2 x 2 matrix A. And rank parameter k = 1. Then say initially A_1,1 = 1, A_2,2 = 0. My rank-1 approximation to A (i.e. truncated rank-1 SVD) will be [1,0,0,0]. Now say I repeatedly make rank-1 updates where I increment A_2,2 by .5. Then every time I make a rank-1 update and then truncate back to rank-1, I will truncate A back to [1,0,0,0]. This will be a very poor approximation to my matrix, which after t updates will have A_11 = 1 and A_22 = t/2.

This issue is known in the literature. For example, it is addressed by work on Frequent Directions sketching — see https://arxiv.org/abs/1501.01711, which performs a different type of truncation that prevent losing large weight of accumulated updates.

Critically, the main regret bound of the paper Theorem 1, must depend then on some sort of accuracy guarantee for TISVD, since the algorithm it applies to uses TISVD. However, if one inspects the proof of Theorem 1, there is no mention of TISVD. From what I understand, it is simply assumed that TISVD does in fact output an optimal rank-k approximation to the accumulated updates (I believe specifically in equation (16)).

It would be very helpful if the authors could address the above concerns during the author response period and correct me if I am mistaken.

**Strengths:**

See full review.

**Weaknesses:**

See full review.

**Questions:**

I have starred questions/comments that I think are more important and would be helpful to see in the rebuttal.
Questions/Comments:
- I don’t understand the premise in the first paragraph. Why does OGD only use a linear combination of the feature vectors? This is only if the gradient is a linear combination of these vectors right? It would depend on the function being optimized no? OGD could be applied to highly non-linear models, such as neural networks. Are you focusing specifically in linear regression and kernel ridge regression here? It is unclear.
-** In the discussion of the work of Calandriello et al. it says ‘rendering a significant cost of updates’. I didn’t follow this? Why is this? Isn’t the goal of that line of work exactly to make the cost scale linearly in the number of data points seen?
-** In abstract and intro it is not clear what ‘the budget’ is. Relatedly I don’t understand why the work of Calandriello has a quadratic scaling. Can this be made more concrete?
- In the problem set up, can what class does f_t come from? Presumably it can’t be just and function mapping inputs to labels. Are you assuming that it is linear in the RKHS?
-** In the classic Nystrom method of Williams and Seeger, the approximation of the for KS(S^TKS)^+ S^K where S is a sampling matrix. I.e. U = (S^TKS)^+. See equation (10) in Williams and Seeger. U is very efficient to compute since S^T K S is just a small principal submatrix of the kernel matrix. Further, this approximation is optimal in the trace norm.
- From what I can tell the approach described in (1), which optimizes U in the Frobenius norm, was not discussed in the Williams and Seeger paper, and is much more computationally expensive as it requires computing (KS)^+. Why is this variant being considered? Why does classic Nystrom not suffice? The approach of (2) seems reasonable, but why use it over classic Nystrom? Is it more accurate in some settings? Some justification needs to be given here.
-** Why in 5 are you using a rank k SVD. Is the goal to further reduce dimension? If so, why?  Again — this needs to be justified as this seems to be a key computational step tackled by the presented approach.
- Incremental SVD algorithms/algorithms for SVD with low-rank updates are widely studied. See e.g. https://www.merl.com/publications/docs/TR2006-059.pdf. More work should be cited here and a more clear comparison made.
- The discussion of the randomized SVD in Appendix D might be inaccurate?. Assuming you are maintaining a reduced rank-k SVD of the s x s matrix, then this matrix can be multiplied by in O(s*k) time, not O(s^2) time. By multiplying by each factor sequentially. Thus, the runtime here I believe should be O(s*k^2+k^3).
- In Theorem 1, the coherence is undefined. This has multiple different definitions in the literature so needs to be defined.
-** Table 1 doesn’t make sense to me. If one wants to achieve a faster algorithm, why not just set B = 0. Or just design an algorithm that actually looks at just a small fraction of the budget, giving any runtime desirable. In order for this table to sensical, there must be some notion of how the budget and the regret bound relate. Also as mentioned earlier — the term ‘budget’ is never defined. It is not clear what it means.

---

> ### Author Response · Authors · 2023-11-18
>
> **Response to Reviewer SDsQ**:
>
> Thanks for your detailed review of our work!
>
> **Reply to Questions**:
>
> Thank you to the reviewers for their valuable suggestions on TISVD. It is important to clarify that our article primarily focuses on enhancing existing second-order online kernel learning methods. Our emphasis is on both the regret guarantee of the algorithm and its performance, particularly on large-scale datasets.
> In reference to Calandriello's work, which involves maintaining an approximate kernel matrix with a size of $B$ through online sampling, it is noteworthy that its feature mapping dimension is also $B$. This characteristic results in a quadratic order of second-order updates, influencing the update cost. Additionally, it is essential to highlight that once the approximate kernel matrix reaches the specified budget, its feature mapping remains unchanged, rendering it susceptible to distribution drift.
>
> The concept of a "budget" is widely employed in online kernel learning. Given the escalating size of the kernel matrix over time, leading to uncontrollable time complexity, setting a sufficient budget becomes imperative to constrain the matrix size. Reference [1] presents a table detailing update times, and we believe that Table 1 effectively illustrates the comparison of update times and regrets across various online kernel learning algorithms.
>
> [1]  Jing Lu, Steven CH Hoi, Jialei Wang, Peilin Zhao, and Zhi-Yong Liu. Large scale online kernel learning. Journal of Machine Learning Research, 17(47):1, 2016.

---

> > ### Comment · Reviewer_SDsQ · 2023-11-18
> >
> > Thanks for the response. Just to clarify though — the regret guarantee of Algo 1 is incorrect as stated, correct? Since it assumes that TISVD in fact returns an optimal low rank approximation, which it does not in general.

---

> > > ### Author Response · Authors · 2023-11-18
> > >
> > > Yes, we will make modifications to Theorem 1. Thank you for your suggestions.

---

### Official Review · Reviewer_Rsps · 2023-11-04

**Soundness:** 3 good
**Presentation:** 2 fair
**Contribution:** 3 good
**Rating:** 5
**Confidence:** 4

**Summary:**

This paper proposes a novel variant of second-order online sketched kernel learning with a reduced complexity in time and memory. The approach exploits a key ingredient, the so-called Truncated Incremental SV,  when updating the approximated sketched feature map.
TiSVD allows to incrementally modify a SVD matrix (when new data arrives) by directly modifying its different components and consequently update at a lower price the approximated sketched feature map. This trick is used in association with second order corrections of the weight w_t. The contribution is simple to understand when reading appendix C and it nicely completes the literature. A regret analysis is developed with an assumption about directional curvature of the  Lipchitz continuous loss (no strong convexity is required) additionally to classic assumptions about polynomial weight decay of the base kernel. Experimental results include comparisons between this algorithm and competitors with fixed budget (nb of streaming data with non-zero loss for which corrections have to be made). Moreover, FORK is also empirically shown to be robust against adversarial attacks.

**Strengths:**

Overall, the paper is well written. Its strength is to combine a technical contribution with theoretical and empirical back up. The work, well executed, appears as the first one to reduce the complexity to a linear order of a budget and therefore, provides a useful contribution to the ML community.

TiSVD is original up to my knowledge. However, originality is somehow limited in the sense that the contribution consists essentially in introducing TiSVD in the algorithm for updating the approximated sketched feature map while the overall algorithm already pre-exists.

**Weaknesses:**

Overall, my main concern is that this work is very incremental while bringing a new component in second-order online sketched kernel algorithms. It certainly deserves a publication but I am not completely convinced that it should be in a venue like ICLR.

The paper presents some flaws in its presentation that can be easily solved.
- Related works are not presented in sufficient details so that one can measure the importance of the contribution.
- a few lines are needed to close the loop between the description of TiSVD and its usage in the algorithm, too much material is delayed to the supplements
- more importantly, Assumption 3 is badly written: 1/ it is written as a definition while it is an assumption about S_p. 2/ It contains a big typo, the equation as presented here seems to me a non-sense. I assume that the event on which the probability is computed should be $\|  B_iA_i  - B_I S_p S_p^T A_i \|^2 > Constant \| A_i \|^2 \|B_I \|^2$ with the right notations...

**Questions:**

- Please following the previous remarks, comment on novelty of the paper
- can you come back to assumption 3 and confirm the typo and propose a clean version of it ?
- discuss the choice of k (in empirical study, it is fixed for comparison which is fine)
- Adversarial attacks are empirically studied. I did not catch why the algorithm should be robust to such attacks. Is a theoretical guarantee is reachable ?
- finally, overcoming concept drifting is also a claim of the paper but I don't really see a discussion about that.

The reviewers have not answered to my concerns.

---

> ### Author Response · Authors · 2023-11-18
>
> **Response to Reviewer Rsps**:
>
> Thanks for your detailed review of our work!
>
> **Reply to Weaknesses**:
>
> Regarding assumption 3, which is a typo, we will modify it as follows:
>
> $\mathrm{Pr} (  \|  B_i  A_i -  B_i {S_p} { S_p}^{\top}  A_i   \|_{F}^2 > 2\| || B_i ||_F^2 || A_i ||_F^2 / (\delta_i s_p)  ) \leq \delta_i$

---

### Official Review · Reviewer_XAux · 2023-11-04

**Soundness:** 2 fair
**Presentation:** 2 fair
**Contribution:** 1 poor
**Rating:** 3
**Confidence:** 4

**Summary:**

This paper proposes a fast second-order approach, called FORKS, for online kernel learning. The FORKS method leverages incremental sketching to reduce the computational cost. The authors provide both theoretical and empirical analysis for the proposed method.

**Strengths:**

The proposed method requires less computational cost than existing second-order approaches for online kernel learning.

**Weaknesses:**

1. The novelty of this paper is limited. The TISVD algorithm, which claimed to be novel, seems the same as the method in (Brand 2006). The incremental maintenance of sketch is almost same as the method in (Zhang & Liao, 2019).
2. This paper does not seems to consider the approximation error caused by truncated SVD.

**Questions:**

see weakness

---

> ### Author Response · Authors · 2023-11-18
>
> **Response to Reviewer XAux**:
>
> Thanks for your detailed review of our work!
>
> **Reply to Weaknesses**:
>
> Here we'd like to highlight the novelty and challenge of adapting the first-order method to the second-order method as follows:
>
> *    The prevailing truncated SVD method proves to be inefficient when employed within the incremental sketching framework, primarily due to its inadequacy in terms of incremental maintenance. In contrast, our proposed TISVD dramatically expedites online learning algorithms while avoiding the undue loss of information.
>
> *    In contrast to first-order online kernel learning algorithms, second-order methods exhibit a heightened sensitivity to initial conditions owing to their rapid convergence characteristics. To mitigate potential adversarial effects, we introduce a reset step to establish a stable starting point. Experimental validation underscores that the reset method significantly enhances the performance of second-order algorithms in the realm of online learning.
>
> *    The current landscape of online kernel learning research notably lacks substantial testing on large-scale datasets. Additionally, existing studies predominantly rely on serialized datasets derived from offline sources in the domain of online kernel learning. However, these datasets lack the dynamic concept drifting seen in datasets from streaming applications, rendering them insufficient to adequately demonstrate the advantages of online learning paradigms. We verify the efficiency of our method by conducting experiments on the dataset *KuaiRec* (Table 5), which, to the best of our knowledge, is currently the largest real-world streaming dataset used to test online kernel learning algorithms.
>
> We agree that the the approximation error caused by truncated SVD should be considered in the theory, We will make it the focus of the modification.

---

> > ### Comment · Reviewer_XAux · 2023-11-22
> >
> > Thank you for the response. After reading the rebuttal and other reviews, I will keep my score.

---

### Meta-Review · Area_Chair_vM5q · 2023-12-11

**Metareview:**

The paper proposes a sketched second-order approach to online kernel learning, and establishes that it has logarithmic regret similar to second-order approaches, while being more computationally efficient due to the use of incremental low-rank approximations. Although experimental evidence is provided to show that the proposed algorithm has a favorable trade-off between accuracy and computation, the novelty of the method relative both to prior approaches to truncated incremental SVD algorithms and applications of Nystrom sketching to online kernel leanring is limited. The regret analysis is also inaccurate, as it does not account for the fact that the incremental low-rank approximation used in the algorithm may not track the true exact low-rank approximation.

**Justification For Why Not Higher Score:**

The novelty is limited and the regret analysis is inaccurate.

**Justification For Why Not Lower Score:**

N/A

---

### Decision · Program_Chairs · 2024-01-16

Reject